# MolReFlect: Towards In-Context Fine-grained Alignments between Molecules and Texts

## Abstract

Molecule discovery is a pivotal research field, impacting everything from the medicines we take to the materials we use. Recently, Large Language Models (LLMs) have been widely adopted in molecule understanding and generation, yet the alignments between molecules and their corresponding captions remain a significant challenge. Previous endeavours often treat the molecule as a general SMILES string or molecular graph, neglecting the fine-grained alignments between the molecular sub-structures and the descriptive textual phrases, which are crucial for accurate and explainable predictions. In this case, we introduce Mol-ReFlect, a novel teacher-student framework designed to contextually perform the molecule-caption alignments in a fine-grained way. Our approach initially leverages a larger teacher LLM to label the detailed alignments by directly extracting critical phrases from molecule captions or SMILES strings and implying them to corresponding sub-structures or characteristics. To refine these alignments, we propose In-Context Selective Reflection, which retrieves previous extraction results as context examples for teacher LLM to reflect and lets a smaller student LLM select from in-context reflection and previous extraction results. Finally, we enhance the learning process of the student LLM through Chain-of-Thought In-Context Molecule Tuning, integrating the fine-grained alignments and the reasoning processes within the Chain-of-Thought format. Our experimental results demonstrate that MolReFlect enables LLMs like Mistral-7B to significantly outperform the previous baselines, achieving SOTA performance on the ChEBI-20 dataset. This advancement not only enhances the generative capabilities of LLMs in the molecule-caption translation task, but also contributes to a more explainable framework.

## 1 Introduction

Molecules are the fundamental units of matter, which normally consist of atoms held together by chemical bonds. In various chemical and biological processes, molecules play a critical role in participating in reactions (Grozinger & Schreiber, 2002), transmitting signals (Raymo & Giordani, 2001), and maintaining the structure and function of living organisms (Konieczny et al., 2023). It is important to study molecules and their properties, which could benefit a wide range of fields, including Pharmacology (Keiser et al., 2010), Agriculture (Twyman et al., 2003; Basaran & Rodríguez-Cerezo, 2008), Material science (Higuchi et al., 2023), and Environmental Ecology (Nguyen et al., 2017; Valavanidis et al., 2006).

As molecules can be represented by textual systems like SMILES (Weininger, 1988) and SELFIES (Krenn et al., 2020), it is natural to adopt Large Language Models (LLMs) in molecule-related tasks (Zhang et al., 2024). Specifically, LLMs could predict the molecular properties based on the SMILES or SELFIES representations and generate molecules with desired properties, making them helpful assistants for chemists. Correspondingly, Edwards et al. (2022) propose the molecule-caption translation task to bridge the gap between molecular and natural language space, which includes molecule captioning (Mol2Cap) and text-based de novo molecule generation (Cap2Mol). In addition to text, several multi-modal methods, like MoMu (Su et al., 2022) and MolCA (Liu et al.,

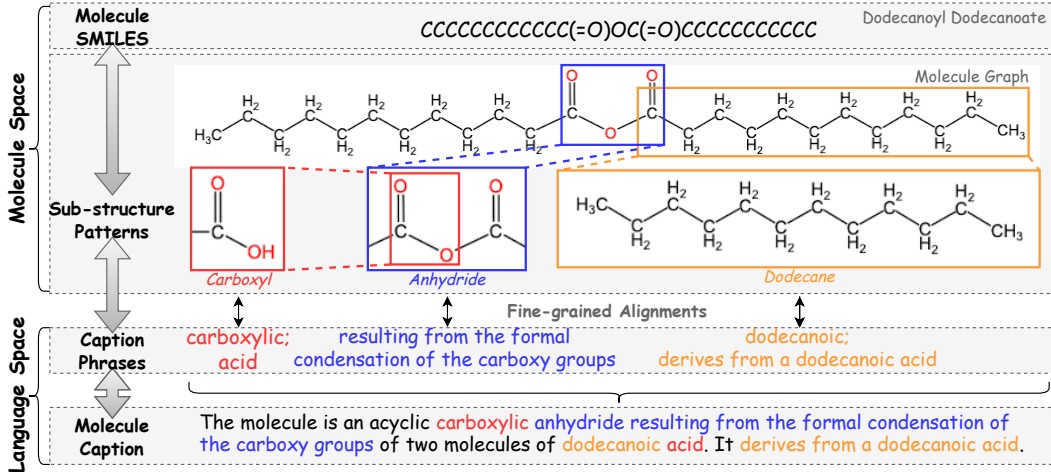

Figure 1: An illustration of the alignments between the molecular space and the language space. The sub-structure patterns are highlighted with colours, and their corresponding caption phrases are also coloured with the same colours to signify the alignments. Here, the molecule *Dodecanoyl Dodecanoate* (*CCCCCCCCCCCC(=O)OC(=O)CCCCCCCCCCC*) is the reaction production of two dodecanoic acids. Thus, it has an anhydride group, and there are 12 carbon atoms on each side of the central oxygen atom.

2023), have been explored by introducing extra information from different modalities to the LLMs. However, challenges still exist in the alignments between molecules and texts.

Current methods typically require an extra modality alignment stage, which suffers from the lack of high-quality molecule-caption pairs. Furthermore, these methods still treat the whole molecule as a general textual string or molecular graph, neglecting the granularity of alignments and the explainability of their methods. Specifically, sub-structures in the molecule, such as functional groups, exactly determine the characteristics of the molecule described in the molecule caption. Similarly, the characteristics described in the molecule caption also directly refer to specific sub-structures of the molecule. For example, as shown in Figure 1, the molecule *Dodecanoyl Dodecanoate* is the reaction production of the formal condensation of two dodecanoic acids, which turns two carboxyls (*RC(=O)OH*) into an anhydride (*RC(=O)OC(=O)R*). Thus, it has an anhydride group and there are 12 carbon atoms on each side of the central oxygen atom. If LLMs could notice these patterns, they are more likely to make accurate predictions. In this case, it is crucial to pay attention to the fine-grained alignments between molecules and texts by focusing on decisive sub-structures and caption phrases. Nevertheless, few works have paid attention to refining the granularity of alignments between molecular sub-structures and their corresponding descriptive texts, as such fine-grained alignments often require domain experts for the labelling, which is both costly and time-consuming.

To resolve the above challenges, we propose **MolReFlect**, a teacher-student framework inspired by reflection tuning (Li et al., 2024b), which enables a larger teacher LLM to collaborate with a smaller student LLM for in-context fine-grained alignments in the molecule-caption translation task. The detailed model structure is shown in Figure 3. Generally, MolReFlect includes three stages: **Zero-shot Alignment Extraction**, **In-Context Selective Reflection**, and **Chain-of-Thought In-Context Molecule Tuning (CoT-ICMT)**. Initially, the larger teacher LLM generates zero-shot alignments by extracting important phrases from the molecule SMILES representations or molecule captions and implies them to corresponding characteristics or sub-structure patterns in a zero-shot manner. To improve the quality of the alignments, we further introduce In-Context Selective Reflection, which first retrieves similar samples and their corresponding zero-shot alignments as in-context few-shot examples so that the teacher LLM can reflect on them and then refine its responses. Following this, the student LLM selects between the zero-shot alignments and reflected alignments with lower perplexities to ensure that they could understand the knowledge taught by the teacher LLM and further relieve the noises in the alignments. Finally, to help the student LLM better learn from

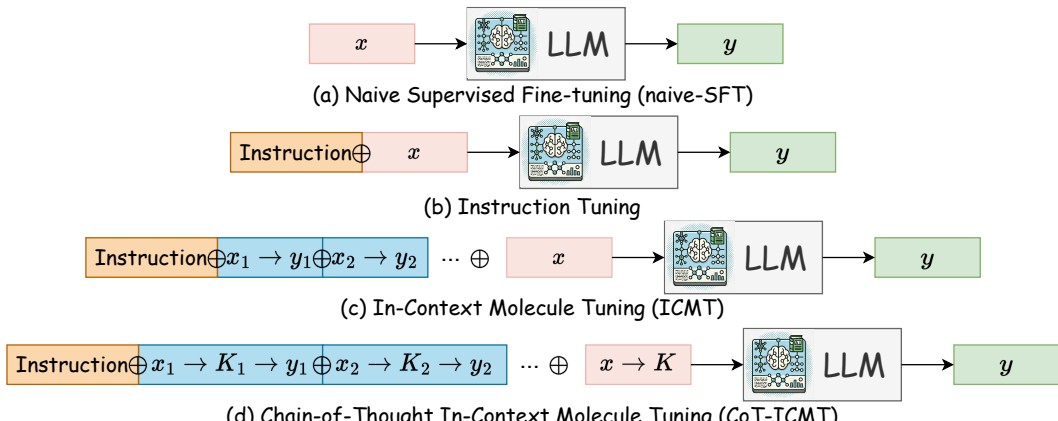

Figure 2: Comparisons of four different fine-tuning paradigms, including (a) Naive Supervised Fine-tuning (naive-SFT), (b) Instruction Tuning (Wei et al., 2021), (c) In-Context Molecule Tuning (ICMT) (Li et al., 2024a), and (d) our proposed Chain-of-Thought In-Context Molecule Tuning (CoT-ICMT).

the fine-grained alignments, we develop a new fine-tuning paradigm, Chain-of-Thought In-Context Molecule Tuning (CoT-ICMT). By reformatting the context examples within a thought chain of $input \rightarrow alignments \rightarrow target$, the reasoning capabilities of LLMs can be better utilized.

To verify the effectiveness of our method and study the mechanisms behind MolReFlect, we design a series of experiments on the ChEBI-20 dataset (Edwards et al., 2022). Experimental results have shown that our method achieves state-of-the-art (SOTA) performance against all the baseline methods in both the Mol2Cap and Cap2Mol tasks. Meanwhile, the ablation studies also demonstrate the effectiveness and mechanism of different stages in MolReFlect. Furthermore, detailed case studies are provided in Appendix C to explain how the fine-grained alignments between molecules and texts improve the overall performance on the molecule-caption translation task with real cases. To summarize, our contributions mainly lie in:

- MolReFlect explores the fine-grained alignments between molecules and texts in a human-free manner. Our method can work with general LLMs without domain-specific pre-training, providing a new solution to relieve the data hunger in the biochemical field.
- By integrating fine-grained alignments into the fine-tuning process of LLMs in the molecule-caption translation task, MolReFlect contributes to a more explainable framework, helping LLMs better understand the translation process between molecules and texts.
- MolReFlect achieves the SOTA performance in the molecule-caption translation task without introducing extra modalities and intricate structures, further demonstrating the importance of in-context fine-grained alignments between molecules and texts.

## 2 PRELIMINARIES

Initially, we explain the differences between three previous fine-tuning paradigms illustrated in Figure 2 (a-c), including Naive-Supervised Fine-tuning, Instruction Tuning (Wei et al., 2021), and In-context Molecule Tuning (Li et al., 2024a). Generally, given an LLM and its parameters $\theta$, supposing that the training set is $D$ and $(x, y) \in D$ denotes a molecule-caption pair from the training set, the LLM ought to generate the response $y \sim p_\theta(.|x)$ based on the input text $x$. Notably, in this paper, $x$ refers to both the input molecule and input caption, while $y$ refers to the corresponding target caption and target molecule. Naive supervised fine-tuning (naive-SFT) learns the mapping from input to target $x \rightarrow y$ directly. Accordingly, the loss function of naive-SFT could be represented as follows:

$$L^{nft}(\theta) = \sum_{(x,y) \in D} [-\log p_\theta(y|x)].$$
(1)

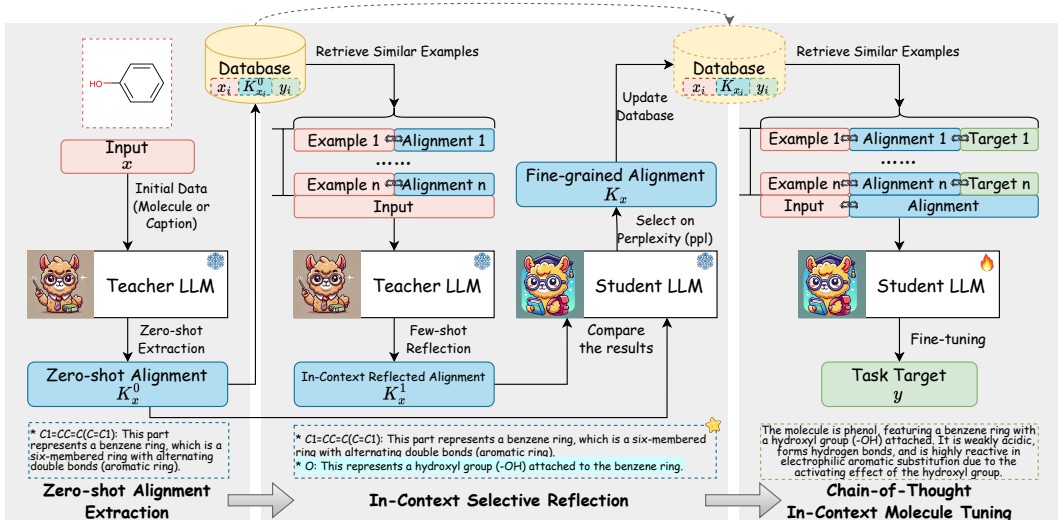

Figure 3: The overall framework of MolReFlect.

Different from naive-SFT, Instruction Tuning (Wei et al., 2021) introduces instructions to guide the generation of LLMs. Normally, instructions contain task-related information such as role identification and additional knowledge. Formally, given the task instruction $I$, the loss function of Instruction Tuning can be denoted as:

$$L^{it}(\theta) = \sum_{(x,y)\in D} \left[ -\log p_\theta(y|x, I) \right]. \tag{2}$$

Inspired by In-Context Tuning (Chen et al., 2022), Li et al. (2024a) take a step further and propose In-Context Molecule Tuning (ICMT) as a crucial stage of In-Context Molecule Adaptation (ICMA), which introduces n similar molecule-caption examples $\{(x_i, y_i)\}_{i=1}^n \subset D$. Therefore, the LLM will make predictions based on the text content $C_{x\to y} = \{\mathcal{P}(x_i, y_i)\}_{i=1}^n$ and the mappings $F_{x\to y} = \{f_i := x_i \to y_i\}_{i=1}^n$ behind the context examples, where $\mathcal{P}$ denotes the prompt template. Thus, as illustrated in Figure 2 (c), the loss function of ICMT can be written as:

$$L^{icmt}(\theta) = \sum_{(x,y)\in D} \left[ -\log p_\theta(y|x, [C_{x\to y}, F_{x\to y}], I) \right], \tag{3}$$

## 3 MOLREFLECT

In this section, we introduce the MolReFlect framework. As depicted in Figure 3, MolReFlect employs a teacher-student architecture, where an advanced (larger) language model serves as the teacher, and a less sophisticated (smaller) language model acts as the student. The teacher LLM collaborates with the student LLM to fine-grain the in-context alignment between molecules and texts, thereby enhancing the overall efficacy in the molecule-caption translation task. The MolReFlect framework is organized into three principal stages: Zero-Shot Alignment Extraction, In-Context Selective Reflection, and CoT-ICMT. We proceed to elaborate on each of these stages in sequence.

### 3.1 ZERO-SHOT ALIGNMENT EXTRACTION

Previously, the molecule-caption translation task treats a molecule as a general SMILES string $m$ and tries to let LLMs learn the direct mappings $m \leftrightarrow c$ between the molecule SMILES string $m$ and the textual caption $c$. To refine the alignments, several multi-modal methods like MolCA (Liu et al., 2023) have been proposed to incorporate molecule graph information $g_m$ and learn the direct mapping $(m, g_m) \to c$ for the Mol2Cap task. Nevertheless, these methods still treat the molecule as a

general SMILES sequence or molecular graph while ignoring the significance of detailed molecular sub-structures.

Instead of directly learning the mappings from molecules to captions, MolReFlect aims to extract fine-grained alignments $K$ between the molecule SMILES strings and molecule captions, thereby learning the mapping chains $m \to K \to c$ and $c \to K \to m$. Typically, the fine-grained alignments should be labeled by professional chemists, which is not only challenging but also financially prohibitive. As a result, LLMs have emerged as a viable alternative due to their advanced reasoning capabilities and a certain degree of chemical knowledge. Within MolReFlect, we have developed a zero-shot prompting strategy to empower the teacher LLM to engage in chain-of-thought (CoT) reasoning (Wei et al., 2022). This allows the teacher LLM to distill critical fragments from the molecule SMILES representations or captions, offering implications to their corresponding properties or sub-structure patterns. Formally, we have:

$$K_c^0 = p_{\theta_T}(.|c, I), \ K_m^0 = p_{\theta_T}(.|m, I), \tag{4}$$

where $\theta_T$ represents the parameters of the larger teacher LLM, $I$ is the CoT instruction, and $K_c^0$ and $K_m^0$ signify the alignments extracted in a zero-shot manner from the molecule caption and SMILES string, respectively.

## 3.2 In-Context Selective Reflection

Despite the powerful capabilities of LLMs, they can still generate answers with hallucinations (Yao et al., 2023). Their knowledge of chemistry is also limited due to the absence of domain pre-training on chemical corpora, which can introduce noises into the zero-shot alignments. To mitigate these potential noises and enhance the quality of zero-shot alignments, we propose a strategy that allows the larger teacher LLM to self-reflect on the zero-shot extraction results through in-context few-shot learning, where the previous zero-shot alignments are retrieved by similarity and serve as context examples for reflection. From the perspective of the molecular similarity principle, we do not calculate the similarity among the fine-grained alignments but follow the retrieval strategy adopted in Li et al. (2024a). For caption-based retrieval, we calculate the caption similarities based on the BM25 algorithm (Robertson et al., 2009) and retrieve top $n$ similar captions $\{c_1, c_2, ..., c_n\}$ ranked by the BM25 scores and their corresponding zero-shot alignments $\{K_{c_1}^0, K_{c_2}^0, ..., K_{c_n}^0\}$ for the input caption $c$ to form the context examples $C_c$:

$$C_c = \{(c_1, K_{c_1}^0), (c_2, K_{c_2}^0), ..., (c_n, K_{c_n}^0)\} \tag{5}$$

Similarly, for molecule retrieval, we employ a pre-trained Mole-BERT (Xia et al., 2022) as the graph encoder and calculate the cosine similarities between the molecule graph embeddings. Top $n$ similar molecules $\{m_1, m_2, ..., m_n\}$ and their corresponding zero-shot alignments $\{K_{m_1}^0, K_{m_2}^0, ..., K_{m_n}^0\}$ are retrieved for the input molecule $m$ as the context examples $C_m$:

$$C_m = \{(m_1, K_{m_1}^0), (m_2, K_{m_2}^0), ..., (m_n, K_{m_n}^0)\} \tag{6}$$

Based on the input $c$ or $m$, context examples $C_c$ or $C_m$, and instruction $I$, we could obtain the in-context reflected alignments $K_c^1$ or $K_m^1$ through the teacher LLM. Notably, the zero-shot alignments of the current input are not wrapped into the context to prevent the LLM from directly repeating it, and maintain consistent prompt formats across all instances:

$$K_c^1 = p_{\theta_T}(.|c, C_c, I), \ K_m^1 = p_{\theta_T}(.|m, C_m, I), \tag{7}$$

However, the context examples might also introduce noises that could misguide the reflection process, potentially leading to a decline in the quality of the reflected alignments $K^1$ compared to the zero-shot alignments $K^0$. Furthermore, the alignments generated by the teacher LLM can sometimes be too complex for the smaller student LLM to comprehend. Therefore, choosing the superior one between $K^0$ and $K^1$ is essential. To avoid possible information leaks, an unsupervised metric is required for selection. Specifically, we adopt the perplexity ppl as the metric from the information theory perspective:

$$\text{ppl}(K_x, x) = \log\left[-p_{\theta_S}(K_x|x)\right], \tag{8}$$

where $x$ is the input, $K_x$ denotes the corresponding alignments, and $\theta_S$ is the original parameters of the smaller student LLM. Higher perplexity scores suggest the presence of information that conflicts

with the existing knowledge of LLMs. Therefore, the student LLM used for perplexity calculation is better to have some chemical knowledge like Galactiva-125M (Taylor et al., 2022) and can be different from the student LLM used for CoT-ICMT. Between the zero-shot alignment and the in-context reflected alignment, the one with lower perplexity will be selected:

$$K_c = \begin{cases} K_c^0 & \text{if } \mathrm{ppl}(K_c^0, c) < \mathrm{ppl}(K_c^1, c), \\ K_c^1 & \text{elsewise}, \end{cases} \tag{9}$$

$$K_m = \begin{cases} K_m^0 & \text{if } \mathrm{ppl}(K_m^0, m) < \mathrm{ppl}(K_m^1, m), \\ K_m^1 & \text{elsewise}, \end{cases} \tag{10}$$

### 3.3 CHAIN-OF-THOUGHT IN-CONTEXT MOLECULE TUNING

While it is technically possible to leverage fine-grained alignments as contexts to allow the larger teacher LLM to generate final predictions directly in a CoT manner, the teacher LLM still lacks specialized pre-training on chemical corpora and is unfamiliar with the specific output distribution of the dataset. Consequently, directly querying the larger teacher LLM for final generations usually leads to unsatisfactory results. Furthermore, the cost of directly fine-tuning the larger teacher LLM is prohibitively high, making it unaffordable for most institutions. Instead, we fine-tune the smaller student LLM to learn from the fine-grained alignments provided by the larger teacher LLM. Notably, in this phase, we prioritize the reasoning capabilities of the student LLM over their knowledge of chemistry, so it can differ from the student LLM used to calculate perplexity.

In contrast to In-Context Molecule Tuning (Li et al., 2024a), CoT-ICMT organizes the fine-grained alignments of both the input $x$ and the context examples $C_x$ into the CoT format. This CoT format empowers LLMs to learn from the fine-grained alignments and the reasoning processes behind the context examples, thereby facilitating more explainable training. During the process of CoT-ICMT, top-$n$ similar examples are retrieved via the same retrieval strategies mentioned in Section 3.2 and then organized into the context with the CoT format to fine-tune the parameters of the smaller student LLM. Formally, similar to Eq. 3, the loss function can be represented as follows:

$$L^{cot-icmt}(\theta) = \sum_{(x,y) \in D} \left[ -\log p_\theta(y | x, K_x, [C_{x \to K_x \to y}, F_{x \to K_x \to y}], I) \right], \tag{11}$$

where $K_x$ denotes the fine-grained alignments of input $x$, $C_{x \to K_x \to y} = \{\mathcal{P}(x_i, K_{x_i}, y_i)\}_{i=1}^n$ represents the text content of context examples organized by the CoT format prompt $\mathcal{P}$, and $F_{x \to K_x \to y} = \{f_i := x_i \to K_{x_i} \to y_i\}_{i=1}^n$ denotes the mapping chains behind the context examples, which map the original inputs to the fine-grained alignments and then further map the fine-grained alignments to the final targets.

## 4 EXPERIMENTS

In this section, we first present our experiment setups and compare MolReFlect against existing baselines. Then, we conduct a series of ablation experiments to validate our proposed framework, focusing on the following specific research questions: **(RQ1)** Do fine-grained alignments improve the performance in the molecule-caption translation task, and if so, how? **(RQ2)** Why is it necessary to reflect and select between the zero-shot alignments and in-context reflected alignments? **(RQ3)** What is the necessity of adopting a teacher-student framework?

### 4.1 EXPERIMENT SETUPS

**Implementation Details.** For the larger teacher LLM, we adopt the powerful Llama-3-70B-Instruct model (Dubey et al., 2024), as its competitive performance against GPT-4 (Achiam et al., 2023) makes it well-suited for the role of teacher. For the smaller student LLM, we mainly adopt Mistral-7B-Instruct-v0.2 (Mistral-7B for short) (Jiang et al., 2023) for fair comparisons to ICMA (Li et al., 2024a). In this work, we focus on the ChEBI-20 dataset (Edwards et al., 2022) and all the experiments are conducted on Nvidia RTX A6000 and A100 GPUs. Appendix A provides more implementation details and hyper-parameter lists.

**Metrics.** Regarding the evaluation metrics, we adopt the same settings as ICMA. We employ translation metrics for the Mol2Cap task, including BLEU-2,4 scores, ROUGE-1,2,L scores, and METEOR scores. Higher values in these metrics indicate that the generated molecule captions are more aligned with the ground truth. For the Cap2Mol task, we employ a combination of translation and molecule-specific metrics for evaluation, which includes BLEU, Exact Match, Levenshtein, three Molecule Fingerprints scores, and a validity score. Except for the Levenshtein score, where a lower value is preferable, higher scores across these metrics generally signify better model performance.

## 4.2 OVERALL PERFORMANCE COMPARISON

We compare our method with the baseline models across the two sub-tasks of the ChEBI-20 dataset. Specifically, we select MolT5-large (Edwards et al., 2022), MolReGPT (Li et al., 2023a), MolCA (for the Mol2Cap task only) (Liu et al., 2023), BioT5 (Pei et al., 2023), and ICMA (Li et al., 2024a) as the baseline models. Notably, we adopt Mistral-7B as the smaller student LLM in the CoT-ICMT stage of MolReFlect. The overall results are presented in Table 1 for the Mol2Cap task and in Table 2 for the Cap2Mol task. We will proceed to discuss the outcomes for each sub-task individually.

Table 1: Overall performance comparison for the Mol2Cap task on the ChEBI-20 dataset (**Best**, Second Best). Except for MolReGPT, all the other methods involve fine-tuning LLMs on the ChEBI-20 dataset.

| Method | BLEU-2↑ | BLEU-4↑ | ROUGE-1↑ | ROUGE-2↑ | ROUGE-L↑ | METEOR↑ |
|---|---|---|---|---|---|---|
| MolT5-large | 0.594 | 0.508 | 0.654 | 0.510 | 0.594 | 0.614 |
| MolReGPT | 0.607 | 0.525 | 0.634 | 0.476 | 0.562 | 0.610 |
| MolCA | 0.639 | 0.555 | 0.697 | 0.558 | 0.636 | 0.669 |
| BioT5 | 0.635 | 0.556 | 0.692 | 0.559 | 0.633 | 0.656 |
| ICMA | 0.651 | 0.581 | 0.686 | 0.550 | 0.625 | 0.661 |
| **MolReFlect** | **0.676** | **0.608** | **0.703** | **0.571** | **0.644** | **0.680** |

**Mol2Cap Task.** As indicated in Table 1, MolReFlect achieves the top scores across all evaluation metrics. Significantly, with the same backbone model Mistral-7B, MolReFlect obtains a BLEU-2 score of 0.676 and a BLEU-4 score of 0.608, representing improvements of 3.8% and 4.6% over ICMA, while maintaining superior ROUGE scores. In comparison to domain-specific pre-training approaches such as BioT5 and multi-modal strategies like MolCA, MolReFlect still exhibits superior performance using a general-purpose LLM without any extra domain-pre-training or modality alignment stages, thereby underscoring the importance of in-context fine-grained alignments between molecules and texts.

Table 2: Overall performance comparison for the Cap2Mol task on the ChEBI-20 dataset (**Best**, Second Best). Except for MolReGPT, all the other methods involve fine-tuning LLMs on the ChEBI-20 dataset.

| Method | BLEU↑ | EM↑ | Levenshtein↓ | MACCS FTS↑ | RDK FTS↑ | Morgan FTS↑ | Validity↑ |
|---|---|---|---|---|---|---|---|
| MolT5-large | 0.854 | 0.311 | 16.07 | 0.834 | 0.746 | 0.684 | 0.905 |
| MolReGPT | 0.857 | 0.280 | 17.14 | 0.903 | 0.805 | 0.739 | 0.899 |
| BioT5 | 0.867 | 0.413 | 15.10 | 0.886 | 0.801 | 0.734 | **1.000** |
| ICMA | 0.855 | 0.460 | 18.73 | 0.916 | 0.837 | 0.789 | 0.958 |
| **MolReFlect** | **0.903** | **0.510** | **11.84** | **0.929** | **0.860** | **0.813** | 0.977 |

**Cap2Mol Task.** As evidenced in Table 2, MolReFlect also exhibits superior performance in the Cap2Mol task. Compared to previous baselines such as ICMA, MolReFlect achieves a BLEU score of 0.903 and generates a remarkable 51% exact matched molecules while obtaining a lower Levenshtein score. Moreover, MolReFlect achieves the highest molecule fingerprint scores, indicating that the generations are more similar to the ground truths. Only the validity of generations is slightly below the 100% validity of BioT5, as MolReFlect employs the SMILES representation of molecules. However, using SMILES strings offers the advantage of requiring an extension of the tokenizer vocabulary, which preserves the information from pre-training, and this limitation can be addressed

through various sampling and string-filtering strategies. Given the size of the test set, the validity of MolReFlect is quite satisfying, with only 60 incorrect SMILES out of 3300 generations.

Therefore, across both the Mol2Cap and Cap2Mol tasks, MolReFlect consistently demonstrates state-of-the-art or comparable performance, affirming the effectiveness of our approach.

### 4.3 ABLATION STUDY & DISCUSSION

To enable a better understanding of MolReFlect, we conduct a series of ablation studies to resolve the research questions that have been raised for discussion.

**RQ1: Do fine-grained alignments improve the performance in the molecule-caption translation task, and if so, how?**

For this question, we conduct an ablation study on MolReFlect by removing context examples and fine-grained alignments, downgrading MolReFlect to Instruction Tuning and ICMT, respectively. Meanwhile, we also provide the naive-SFT performance of Mistral-7B. The results are presented in Table 3 and Table 4. It is evident that the naive-SFT results are actually unsatisfying as Mistral-7B lacks specific pre-training on chemical corpora. Meanwhile, when only the context examples are removed, the performances drop slightly but attain a BLEU-4 score of 0.539 on the Mol2Cap task and a BLEU score of 0.886 on the Cap2Mol task, demonstrating a significant performance improvement compared to naive-SFT. Notably, in the Cap2Mol task, the exact match score nearly doubles compared to naive-SFT, indicating that the fine-grained alignments indeed convey much molecular structure information to the student LLM. Furthermore, when fine-grained alignments are removed during the fine-tuning phase, the performances drop in both Mol2Cap and Cap2Mol tasks. This suggests that the LLMs are able to learn molecule-text alignments more effectively from the fine-grained alignments in the context examples, leading to better final generations.

We also include several cases in Appendix C and conduct a series of extensive experiments in Appendix B for better explanation. As depicted in Figure 5, the larger teacher LLM can generate preliminary indications towards the final target and even directly figure out the molecular structure in fine-grained alignments. However, some of these indications might be inaccurate. With CoT-ICMT, the smaller student LLM could learn from the input distribution, identify these errors, and correct them in the final generation process. In this case, as illustrated in Figure 4, the output distribution generated by MolReFlect aligns better with the ground truth. Conversely, MolT5 and ICMA fail to achieve this owing to the lack of fine-grained alignments.

Table 3: Ablation analysis of MolReFlect for the Mol2Cap task performance (Mistral-7B-Instruct-v0.2 as backbone). Above: Mistral-7B(naive-SFT) and MolReFlect; Middle: Ablating Context Examples and Fine-grained Alignments; Below: Ablating In-Context Reflection and Selection.

| Method | BLEU-2↑ | BLEU-4↑ | ROUGE-1↑ | ROUGE-2↑ | ROUGE-L↑ | METEOR↑ |
|---|---|---|---|---|---|---|
| Mistral-7B(naive-SFT) | 0.566 | 0.478 | 0.614 | 0.449 | 0.547 | 0.572 |
| **MolReFlect** | **0.676** | **0.608** | **0.703** | **0.571** | **0.644** | **0.680** |
| w/o Context Examples | 0.617 | 0.539 | 0.657 | 0.510 | 0.593 | 0.623 |
| w/o Fine-grained Alignments | 0.651 | 0.581 | 0.686 | 0.550 | 0.625 | 0.661 |
| w/o In-Context Reflection | 0.648 | 0.580 | 0.700(8) | 0.568(3) | 0.640(7) | 0.678 |
| w/o Selection | 0.672 | 0.604 | 0.701(1) | 0.568(1) | 0.640(9) | 0.677 |

**RQ2: Why is it necessary to reflect and select between the zero-shot alignments and in-context reflected alignments?**

To resolve this question, we ablate MolReFlect by removing the in-context reflection and the selection processes, which is equivalent to replacing the fine-grained alignments with zero-shot alignments and in-context reflected alignments, respectively. The details are shown in the last two rows of Table 3 (for the Mol2Cap task) and Table 4 (for the Cap2Mol task). From Table 3, we can observe that the results without in-context reflection lead to sub-optimal performance as the teacher LLM could make mistakes or yield hallucinations, underscoring the necessity of in-context reflection. However, the in-context reflected alignments are not necessarily better than zero-shot alignments, as evidenced by Table 4. Sometimes, the zero-shot alignments of similar molecules/captions may contain more noises, like hallucinations and factual errors, than helpful information and inadvertently

become part of the context in the in-context reflection phase. The inaccuracies could then carry over to the in-context reflected alignments, potentially harming the final performance. In this case, the zero-shot alignments can be more helpful as the context examples do not pollute them. Therefore, choosing between zero-shot alignments and in-context reflection alignments is imperative to ensure the quality of fine-grained alignments.

From the information theory perspective, our objective is to provide LLMs with more helpful information and less noise while rigorously preventing any disclosure of information about the target. Therefore, perplexity, an unsupervised metric, is an ideal criterion for the selection process. Higher perplexity scores suggest the presence of information that conflicts with the existing knowledge of LLMs, making it a reliable indicator for discerning the quality of the generated alignments. In this work, we utilize the Galactica-125M as the student model to calculate perplexity, which is particularly adept at chemical tasks and offers rapid computation. The alignments with the lower perplexity scores are selected as the fine-grained alignments. According to Table 3 and 4, across both the Cap2Mol and Mol2Cap tasks, MolReFlect consistently demonstrates superior performance compared to those without in-context reflection or selection, thereby substantiating the effectiveness of In-Context Selective Reflection and Selection.

Table 4: Ablation analysis of MolReFlect for the Cap2Mol task performance (Mistral-7B-Instruct-v0.2 as backbone). Above: Mistral-7B(naive-SFT) and MolReFlect; Middle: Ablating Context Examples and Fine-grained Alignments; Below: Ablating In-Context Reflection and Selection.

| Method | BLEU↑ | EM↑ | Levenshtein↓ | MACCS FTS↑ | RDK FTS↑ | Morgan FTS↑ | Validity↑ |
|---|---|---|---|---|---|---|---|
| Mistral-7B(naive-SFT) | 0.767 | 0.234 | 27.39 | 0.852 | 0.718 | 0.649 | 0.918 |
| **MolReFlect** | **0.903** | **0.510** | **11.84** | **0.929** | **0.860** | **0.813** | 0.977 |
| w/o Context Examples | 0.886 | 0.430 | 13.99 | 0.916 | 0.828 | 0.775 | 0.981 |
| w/o Fine-grained Alignments | 0.855 | 0.460 | 18.73 | 0.916 | 0.837 | 0.789 | 0.958 |
| w/o In-Context Reflection | 0.900(3) | 0.502 | 11.94 | 0.926 | 0.855 | 0.807 | 0.979 |
| w/o Selection | 0.900(1) | 0.496 | 12.86 | 0.927 | 0.858 | 0.808 | **0.980** |

### RQ3: What is the necessity of adopting a teacher-student framework?

In this part, we address the last research question by removing the student model and completing the tasks using only the teacher LLM (i.e., Llama-3-70B). Since the cost of fine-tuning the teacher LLM is unaffordable for most institutions, we only test the performance of teacher LLM with prompt engineering to avoid modifications of their parameters. Various prompting strategies are implemented to enable the teacher LLM to undertake the molecule-caption translation tasks independently, including direct prompting, chain-of-thought prompting, few-shot prompting, and few-shot chain-of-thought prompting. Notably, in the chain-of-thought and few-shot chain-of-thought prompting, we utilize the fine-grained alignments produced by the teacher LLM itself as context information. The results of these experiments are detailed in Table 5 and 6.

It can be observed that while Llama-3-70B is a powerful LLM, its performance under direct prompting is notably weak, as it is not trained on the ChEBI-20 or a lot of chemical corpora, ensuring that the information of the ChEBI-20 dataset is not leaked in its pre-training stage. In the Mol2Cap task, the chain-of-thought strategy enhances the performance by introducing fine-grained alignments. However, in the Cap2Mol task, the performance declines by 1.05%, indicating that the teacher LLM struggles to filter out the noise inherent in the fine-grained alignments without explicit supervisory signals. Similarly, in the few-shot setting, the fine-grained alignments also do not contribute to a significant performance boost for the teacher LLM. In contrast, the student LLM proves to be indispensable and could benefit from the CoT-ICMT process by enabling a better understanding of molecule-text alignments and identifying the noises behind fine-grained alignments. As shown in Table 3 and 4, the Instruction Tuning (i.e., w/o Context Examples) performance increases by 9.94% and 14.22% in the Mol2Cap and Cap2Mol tasks, respectively, compared to the naive-SFT. This further underscores the necessity of discerning and mitigating noise within the fine-grained alignments, suggesting that LLMs must engage in fine-tuning to learn from the fine-grained alignments effectively. Thus, the teacher-student framework proves to be indispensable. It enables the smaller student LLM to learn from the input distribution, discern noise in the content generated by the teacher, and absorb valuable information to inform the final generation process.

Table 5: Performance comparison of prompting strategies for the teacher LLM (Llama-3-70B-Instruct) to perform the Mol2Cap task independently.

| Method | BLEU-2↑ | BLEU-4↑ | ROUGE-1↑ | ROUGE-2↑ | ROUGE-L↑ | METEOR↑ | AVG IMP |
|---|---|---|---|---|---|---|---|
| Direct Prompting | 0.071 | 0.038 | 0.220 | 0.093 | 0.192 | 0.139 | - |
| Chain-of-Thought | 0.149 | 0.075 | 0.249 | 0.089 | 0.204 | 0.179 | 41.80% |
| Few-shot Prompting | 0.457 | 0.389 | 0.556 | 0.399 | 0.492 | 0.481 | - |
| Few-shot Chain-of-Thought | 0.474 | 0.382 | 0.523 | 0.349 | 0.449 | 0.476 | -4.41% |

Table 6: Performance comparison of prompting strategies for the teacher LLM (Llama-3-70B-Instruct) to perform the Cap2Mol task independently.

| Method | BLEU↑ | EM↑ | Levenshtein↓ | MACCS FTS↑ | RDK FTS↑ | Morgan FTS↑ | Validity↑ | AVG IMP |
|---|---|---|---|---|---|---|---|---|
| Direct Prompting | 0.417 | 0.032 | 46.91 | 0.711 | 0.474 | 0.411 | 0.666 | - |
| Chain-of-Thought | 0.380 | 0.033 | 47.46 | 0.708 | 0.476 | 0.407 | 0.683 | -1.05% |
| Few-shot Prompting | 0.773 | 0.134 | 22.53 | 0.869 | 0.748 | 0.679 | 0.751 | - |
| Few-shot Chain-of-Thought | 0.759 | 0.129 | 23.13 | 0.872 | 0.752 | 0.679 | 0.766 | 0.74% |

## 5 RELATED WORK

LLMs have demonstrated great potential in Molecule Discovery, including molecule understanding (Qian et al., 2023), optimization (Ye et al., 2023), and generation (Irwin et al., 2022). To align molecule representation with natural language texts, the MolT5 study first proposed the molecule-caption translation task, introducing a new dataset, ChEBI-20, with pairs of molecule SMILES representations and their textual captions that describe the structural patterns and chemical properties (Edwards et al., 2021). Subsequent research has intensified the focus on this task, branching out in two primary directions.

On one trajectory, the research leverages the in-context learning capability of LLMs and the similarity principle of molecules to help LLMs learn the molecule-text alignment in context (Li et al., 2023a). Advancing this approach, ICMA has developed In-Context Molecule Tuning (ICMT), significantly enhancing the capabilities of LLMs in the molecule-caption translation task and reducing the reliance on domain-specific pre-training (Li et al., 2024a). Concurrently, the other works involve incorporating additional information from different modalities into LLMs. For instance, MoMu (Su et al., 2022) adopts contrastive learning to align the output distribution of the text encoder with the graph encoder, while the CLIP structure (Radford et al., 2021) is not good at generative tasks. In this case, MolCA (Liu et al., 2023) introduce the 2D molecular graph with a Q-Former structure (Li et al., 2023b) to enhance the performance of LLMs in the molecule captioning task. However, the 2D molecular graphs do not actually bring extra information. as the conversion between the molecule SMILES representation and its molecule graph is lossless. Meanwhile, 3D-MoLM (Li et al., 2024c) adopts the similar Q-Former structure, but introduces 3D molecule information to LLMs. However, the 3D information generated by RDKit (Landrum, 2013) is not accurate enough and is not closely related to the molecule properties described in molecule captions.

## 6 CONCLUSION

In this study, we present MolReFlect, a novel teacher-student framework designed to refine the in-context alignments between molecular sub-structures and their corresponding textual descriptions. MolReFlect comprises three stages: Zero-shot Alignment Extraction, In-Context Selective Reflection, and Chain-of-Thought In-Context Molecule Tuning. Fine-tuned with the fine-grained alignments taught by the teacher LLM, the student LLM could benefit from the detailed alignments between molecules and texts, enhancing the overall performance and contributing to a more explainable framework. Our experimental results reveal that MolReFlect outperforms all existing baselines. Additionally, we also substantiate the superior explainability via comprehensive case studies. We believe this work could inspire future works to focus on the granularity of molecule-text alignments in this promising field.

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

# A   DETAILED EXPERIMENT SETUP

**Completions.** For the larger teacher LLM, we adopt the vllm[1] framework to deploy the int4 quantized llama-3-70B-Instruct on the local devices as OpenAI compatible server[2]. On the other hand, for the smaller student LLM, we utilize the huggingface transformers[3] and Lora adapters (Hu et al., 2021) for the fine-tuning process.

Table 7: Hyper-parameters for the larger teacher LLM.

| Item | Value |
|---|---|
| int4 | True |
| temperature | 0.75 |
| top_p | 0.85 |
| top_k | 40 |
| num_return_sequences | 1 |
| max_new_tokens | 512 |
| number-of-examples | 2 |

Table 8: Hyper-parameters for the smaller student LLM.

| Item | Value |
|---|---|
| macro batch size | 32 |
| micro batch size | 1 |
| steps | 8000 |
| warm-up steps | 1000 |
| cutoff length | 4096 |
| number-of-examples | 2 |
| learning rate | 2e-4 |
| lora_r | 32 |
| lora_alpha | 64 |
| lora_dropout | 0.1 |
| int8 | True |
| fp16 | True |
| temperature | 0.75 |
| top_p | 0.85 |
| top_k | 40 |
| num_return_sequences | 1 |
| max_new_tokens | 512 |

**Hyper-parameters.** For reproduction, we list all the hyper-parameters used in our framework, including Table 7 for the prompting of the teacher LLM and Table 8 for the fine-tuning and testing of the student LLM. Notably, we incorporate $n = 2$ examples in both in-context selective reflection and Chain-of-Thought In-Context Molecule Tuning. Furthermore, the Llama-3-70B-Instruct is int4 quantized to allow inference on a single NVIDIA A6000 GPU for data-parallel acceleration, while the Mistral-7B-Instruct-v0.2 is int8 and fp16 quantized during the fine-tuning process. We keep similar generation parameters for both the teacher LLM and the student LLM.

---

[1] https://github.com/vllm-project/vllm

[2] https://platform.openai.com/docs/guides/chat-completions

[3] https://huggingface.co

# B  EXTENSIVE EXPERIMENTS

## B.1  STATISTICS OF FINE-GRAINED ALIGNMENTS

We evaluate the quality of fine-grained alignments with perplexity and an additional metric, semantic similarity, calculated by sentencebert (Reimers & Gurevych, 2019). As shown in Table 9 and 10, as we select the fine-grained alignments by perplexity, the fine-grained alignments naturally inherit the lowest perplexity score. However, it is interesting to see that for the Mol2Cap task, the lower perplexity even indicates better semantic similarity to some extent, which is crucial for the generation of captions. Meanwhile, in the Cap2Mol task, selecting by lower perplexity also relieves the decreased semantic similarity of the in-context reflected alignments, further justifying our design.

Table 9: Average semantic similarity and perplexity scores of different alignments and the original molecules in the training set for the Mol2Cap task.

| Item | semantic similarity | perplexity |
|---|---|---|
| molecules | 0.2483 | 2.246 |
| zero-shot alignments | 0.4983 | 2.066 |
| in-context reflected alignments | 0.4985 | 2.070 |
| fine-grained alignments | 0.5029 | 1.995 |

Table 10: Average semantic similarity and perplexity scores of different alignments and the original molecule captions in the training set for the Cap2Mol task.

| Item | semantic similarity | perplexity |
|---|---|---|
| captions | 0.2483 | 2.758 |
| zero-shot alignments | 0.2721 | 2.426 |
| in-context reflected alignments | 0.2377 | 2.351 |
| fine-grained alignments | 0.2524 | 2.230 |

## B.2  POTENTIAL IN MOLECULE PROPERTY PREDICTION

Although our work is mainly focused on the molecule-caption translation task, we find its potential in molecule property prediction tasks. Here, we evaluate the MolReFlect performance on the BACE and BBBP tasks (Wu et al., 2018). The results are listed in Table 11. Here, we select Mistral-7B, ICMA(Mistral-7B), and MolReFlect (Mistral-7B) to ensure a fair comparison.

Table 11: ROC-AUC (%) scores of MolReFlect on the BACE and BBBP task from the MoleculeNet dataset (Wu et al., 2018) (**Best**, Second Best).

| Tasks | BACE | BBBP |
|---|---|---|
| Mistral7B | 0.4926 | 0.4829 |
| ICMA | 0.7995 | 0.6775 |
| MolReFlect | **0.8795** | **0.8925** |

The results show that MolReFlect achieves the best performance on the two molecule property prediction tasks, showing the potential in generalizing to molecule property prediction tasks.

### B.3 PubChem Performance

To illustrate the generalization performance of MolReFlect, we conduct extensive experiments on the PubChem dataset (Liu et al., 2023). The results are shown in Table 12 and Table 13.

Table 12: Mol2Cap Performance of MolReFlect on the PubChem dataset (**Best**, Second Best). Here, Mistral-7B serves as the backbone LLM.

| Method | BLEU-2↑ | BLEU-4↑ | ROUGE-1↑ | ROUGE-2↑ | ROUGE-L↑ | METEOR↑ |
|---|---|---|---|---|---|---|
| Mistral-7B | 0.361 | 0.288 | 0.471 | 0.325 | 0.419 | 0.421 |
| MolReFlect w/o CoT-ICMT | 0.369 | 0.297 | 0.482 | 0.342 | 0.433 | 0.431 |
| **MolReFlect** | **0.414** | **0.343** | **0.511** | **0.374** | **0.458** | **0.470** |

Table 13: Cap2Mol Performance of MolReFlect on the PubChem dataset (**Best**, Second Best). Here, Mistral-7B serves as the backbone LLM.

| Method | BLEU↑ | EM↑ | Levenshtein↓ | MACCS FTS↑ | RDK FTS↑ | Morgan FTS↑ | Validity↑ |
|---|---|---|---|---|---|---|---|
| Mistral-7B | 43.84 | 8.2 | 74.16 | 73.08 | 57.72 | 47.19 | 86.6 |
| MolReFlect w/o CoT-ICMT | 74.39 | 14.45 | 30.23 | 79.87 | 66.24 | 56.02 | 95.5 |
| **MolReFlect** | **76.32** | **17.15** | **27.69** | **80.6** | **67.76** | **57.65** | **96.2** |

On both Mol2Cap and Cap2Mol tasks, MolReFlect demonstrates the best performance, significantly boosting the generation quality. Meanwhile, the results also show a similar pattern to the ChEBI-20 dataset, which proves the generalization of MolReFlect.

### B.4 Model Agnosticism

To verify the model agnosticism of MolReFlect, we also conduct experiments on a different student LLM, Llama-3-8B-Instruct. We also remove the context examples and fine-grained alignments for ablation purposes. The results are shown in Table 14 and 15. We could observe similar patterns in Llama-3-8B-Instruct compared to Mistral-7B: MolReFlect still achieves the best performance, and when removing context examples and fine-grained alignments, the performance all drops. Meanwhile, MolReFlect also empowers Llama-3-8B-Instruct to achieve SOTA performance on the ChEBI-20 dataset, further demonstrating the model agnosticism of our method.

Table 14: Mol2Cap Performance of MolReFlect when Llama-3-8B-Instruct serves as the student LLM (**Best**, Second Best). We also compare the performance by removing the context examples and fine-grained alignments for ablation purposes.

| Method | BLEU-2↑ | BLEU-4↑ | ROUGE-1↑ | ROUGE-2↑ | ROUGE-L↑ | METEOR↑ |
|---|---|---|---|---|---|---|
| **MolReFlect** | **0.672** | **0.605** | **0.703** | **0.571** | **0.644** | **0.678** |
| w/o Context Examples | 0.617 | 0.540 | 0.661 | 0.515 | 0.598 | 0.622 |
| w/o Fine-grained Alignments | 0.665 | 0.595 | 0.693 | 0.559 | 0.633 | 0.669 |

### B.5 Output Distribution

We also visualize the output distributions of different methods and the ground truth via sentence-bert embeddings (Reimers & Gurevych, 2019), which are shown in Figure 4. It is evident that the output distributions of MolT5 and ICMA are quite different: the caption distribution of MolT5 is more dense, while the caption distribution of ICMA is more sparse. However, MolReFlect generates a similar output distribution compared to the ground truth, better comprehending the mappings between molecules and texts.

### B.6 Study of Model Robustness

To verify the robustness of MolReFlect, we perform the probing test, following the work of Ganeeva et al. by transforming molecular SMILES into equivalent variants. Specifically, four different rules are applied:

Table 15: Cap2Mol Performance of MolReFlect when Llama-3-8B-Instruct serves as the student LLM (**Best**, Second Best). We also compare the performance by removing the context examples and fine-grained alignments for ablation purposes.

| Method | BLEU↑ | EM↑ | Levenshtein↓ | MACCS FTS↑ | RDK FTS↑ | Morgan FTS↑ | Validity↑ |
|---|---|---|---|---|---|---|---|
| **MolReFlect** | **0.896** | **0.472** | **13.33** | **0.925** | **0.846** | **0.797** | **0.979** |
| w/o Context Examples | 0.864 | 0.395 | 16.13 | 0.904 | 0.815 | 0.754 | 0.964 |
| w/o Fine-grained Alignments | 0.851 | 0.445 | 19.27 | 0.915 | 0.836 | 0.785 | 0.958 |

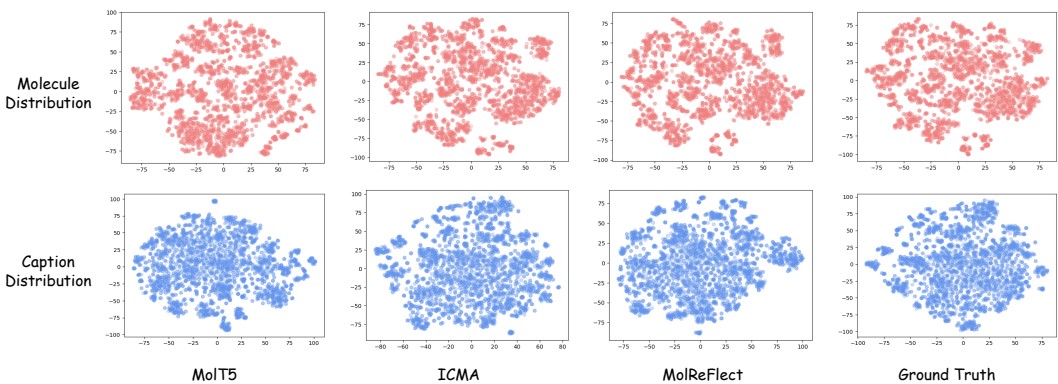

Figure 4: Embedding distributions of molecules and captions.

- **canonicalization**: Transforming a SMILES string into the RDKIT canonical SMILES string.
- **hydrogen**: Adding explicit hydrogen atoms into the SMILES string.
- **kekulization**: Transforming a SMILES string into the kekulized SMILES string.
- **cycles**: Randomly replacing cycle numerical identifiers with other random numbers.

Here, we compare MolReFlect with the following baselines: MolT5-base, MolT5-large Edwards et al. (2022), Text+Chem T5-base, and Text+Chem T5-augm (Christofidellis et al., 2023). The results are shown in Table 16.

Table 16: Results of robustness probing test. The performance on the original test set is labelled as "original". The best performance is **bold** and the second-best performance is underlined.

| Probing Test | MolT5-base | | Text+Chem T5-base | | MolT5-large | | Text+Chem T5-augm | | MolReFlect | |
|---|---|---|---|---|---|---|---|---|---|---|
| | ROUGE-2 | METEOR | ROUGE-2 | METEOR | ROUGE-2 | METEOR | ROUGE-2 | METEOR | ROUGE-2 | METEOR |
| original | 0.481 | 0.583 | 0.498 | 0.604 | 0.510 | 0.614 | 0.543 | 0.648 | **0.571** | **0.680** |
| canonical | 0.315 | 0.450 | 0.381 | 0.515 | 0.390 | 0.532 | 0.377 | 0.514 | **0.416** | **0.543** |
| hydrogen | 0.199 | 0.329 | 0.187 | 0.314 | 0.174 | 0.318 | 0.201 | 0.336 | **0.305** | **0.435** |
| kekulization | 0.333 | 0.475 | 0.413 | 0.574 | 0.405 | 0.546 | 0.410 | 0.546 | **0.443** | **0.569** |
| cycles | 0.417 | 0.540 | 0.483 | 0.600 | **0.566** | 0.603 | 0.4575 | 0.581 | 0.545 | **0.658** |

The results show that although Text+Chem T5-augm achieves better original performance than MolT5-large, the augmentation makes it unrobust to the variance of molecule SMILES. However, MolReFlect not only achieves the highest score on the original test set but also shows the best robustness across the four SMILES variants, further proving the superiority of our MolReFlect.

# C CASE STUDIES

## C.1 FINE-GRAINED ALIGNMENT CASES

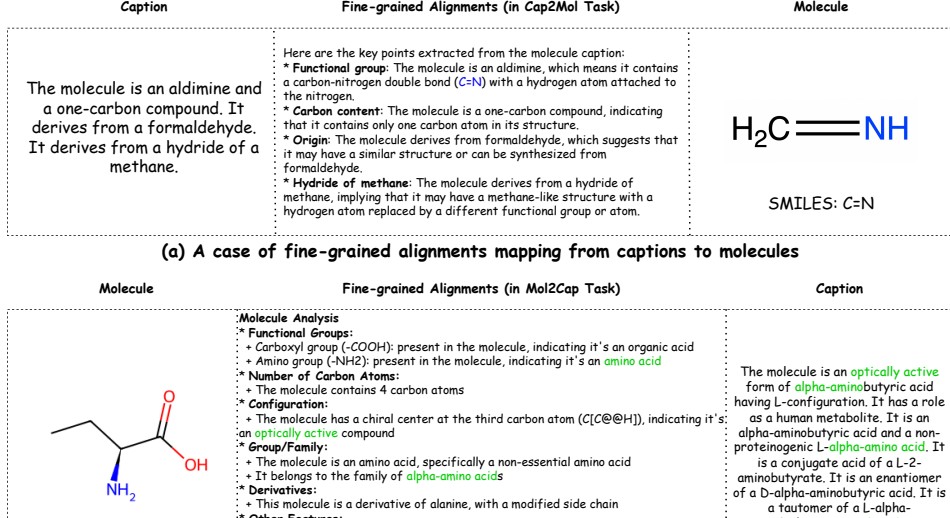

**(a) A case of fine-grained alignments mapping from captions to molecules**

**(b) A case of fine-grained alignments mapping from molecules to captions**

Figure 5: Cases of Fine-grained Alignments. We could observe that the molecule structure and characteristics have already been mentioned and aligned by the fine-grained alignments, which will surely benefit the final generations.

## C.2 CUSTOMIZED CASES

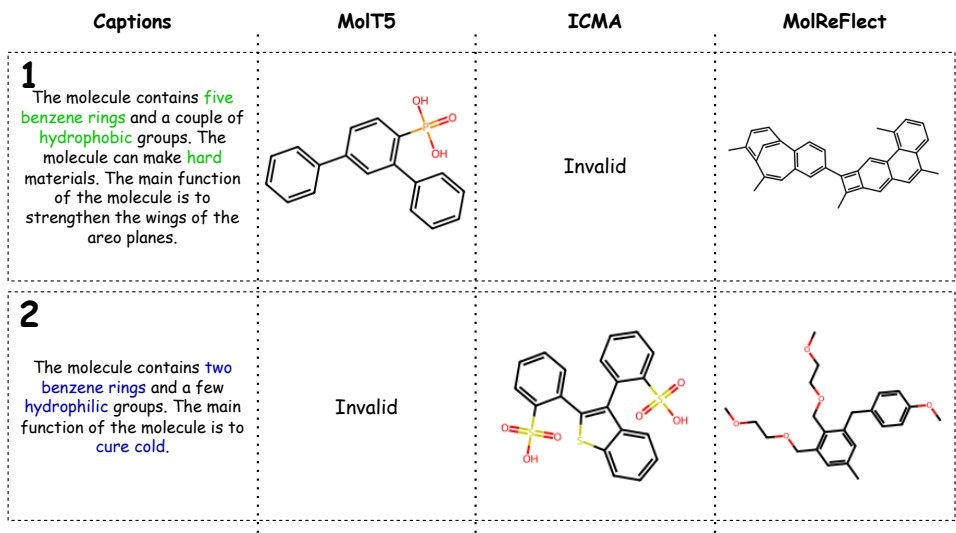

Figure 6: Cases of Customized Examples for the Cap2Mol task. We follow the customized examples in Li et al. (2023a). Obviously, MolReFlect generates correct molecules in general, matching the requirements mentioned in the customized cases, while MolT5 and ICMA fail to meet the requirements.

## C.3 MOL2CAP CASES

| Molecules | MolT5-large | ICMA (Mistral-7B) | MolReFlect (Mistral-7B) | Ground Truth |
|---|---|---|---|---|
| **1** | The molecule is one of three isomeric monoterpenes differing in the positions of their two double bonds (beta- and gamma-terpinene being the others). In alpha-terpinene the double bonds are at the 1- and 3-positions of the p-menthane skeleton. It has a role as a volatile oil component and a plant metabolite. It is a monoterpene and a cyclohexadiene. | The molecule is a cyclic ketone and a ketone body. It has a role as a human metabolite and an angiogenesis inhibitor. It derives from a cyclohexanone and a fumagalone. | The molecule is an alicyclic ketone that is cyclohexanone in which the pro-S hydrogens at positions 2, 3, and 4 are substituted by methoxy, (2E)-6-methylhept-2-en-2-yl, and acetyl groups, respectively. It has a role as an angiogenesis inhibitor. It derives from a fumagalone. | The molecule is an alicyclic ketone that is cyclohexanone in which the pro-S hydrogens at positions 2, 3, and 4 are substituted by methoxy, (2E)-6-methylhept-2-en-2-yl, and acetyl groups, respectively. It has a role as an angiogenesis inhibitor. It derives from a fumagalone. |
| **2** | The molecule is an amino tetrasaccharide in which an alpha-L-fucosyl residue is linked (1->2) to the middle galactosyl residue of an N-acetyl-alpha-D-galactosaminyl-(1->4)-beta-D-galactosyl-(1->4)-N-acetyl-beta-D-glucosamine trisaccharide. It has a role as an epitope. It is a galactosamine oligosaccharide, a glucosamine oligosaccharide and an amino tetrasaccharide. | The molecule is an indolylmethylglucosinolate that is the conjugate base of 4-methoxyglucobrassicin. It is a conjugate base of a 4-methoxyglucobrassicin. | The molecule is an indolylmethylglucosinolate that is the conjugate base of 4-methoxyglucobrassicin, obtained by deprotonation of the sulfo group. It is a conjugate base of a 4-methoxyglucobrassicin. | The molecule is an indolylmethylglucosinolate that is the conjugate base of 4-methoxyglucobrassicin, obtained by deprotonation of the sulfo group. It is a conjugate base of a 4-methoxyglucobrassicin. |
| **3** | The molecule is a branched amino tetrasaccharide comprising N-acetyl-D-galactosamine at the reducing end with an alpha-D-galactosyl-(1->3)-[alpha-L-fucosyl-(1->2)]-beta-D-galactosyl moiety attached at the 3-position. It is an amino tetrasaccharide and a galactosamine oligosaccharide. | The molecule is the dihydrate of the sodium salt of loxoprofen. A prodrug that is rapidly converted to its active trans-alcohol metabolite following oral administration. It has a role as a non-steroidal anti-inflammatory drug, a non-narcotic analgesic, an antipyretic, an EC 1.14.99.1 (prostaglandin-endoperoxide synthase) inhibitor and a prodrug. It contains a loxoprofen sodium. It derives from a loxoprofen. | The molecule is a hydrate that is the dihydrate form of loxoprofen sodium. A prodrug that is rapidly converted to its active trans-alcohol metabolite following oral administration. It has a role as an antipyretic, a non-narcotic analgesic, a non-steroidal anti-inflammatory drug and an EC 1.14.99.1 (prostaglandin-endoperoxide synthase) inhibitor. It contains a loxoprofen sodium. | The molecule is a hydrate that is the dihydrate form of loxoprofen sodium. The parent acid, loxoprofen, is a prodrug that is rapidly converted into its active trans-alcohol metabolite following oral administration. It has a role as a non-steroidal anti-inflammatory drug, a non-narcotic analgesic, an EC 1.14.99.1 (prostaglandin-endoperoxide synthase) inhibitor and an antipyretic. It contains a loxoprofen sodium. |
| **4** | The molecule is a tricyclic triterpenoid isolated from Poria cocos. It has a role as a fungal metabolite. It is a dicarboxylic acid, a secondary alcohol and a tricyclic triterpenoid. | The molecule is a member of the class of formamides that is formamide in which one of the amino hydrogens has been replaced by a butyl group. | The molecule is a member of the class of caprolactams that is epsilon-caprolactam which has been formally dehydrogenated to give the corresponding aldehyde. It has a role as a metabolite. It is an aldehyde and a member of caprolactams. It derives from an epsilon-caprolactam. | The molecule is a member of the class of formamides that is formamide substituted by a butyl group at the N atom. It has a role as a human metabolite. It derives from a formamide. |
| **5** | The molecule is an arabinaric acid. It is a conjugate acid of a L-arabinarate(1-). It is an enantiomer of a D-arabinaric acid. | The molecule is the (S)-enantiomer of 3-sulfolactic acid. It is a conjugate acid of a (S)-3-sulfonatolactate(2-). It is an enantiomer of a (R)-3-sulfolactic acid. | The molecule is the (R)-enantiomer of 3-sulfolactic acid. It is a conjugate acid of a (R)-3-sulfonatolactate(2-). It is an enantiomer of a (S)-3-sulfolactic acid. | The molecule is the (R)-enantiomer of 3-sulfolactic acid. It is a conjugate acid of a (R)-3-sulfolactate. It is an enantiomer of a (S)-3-sulfolactic acid. |
| **6** | The molecule is an optically active form of tyrosine having L-configuration. It has a role as an EC 1.3.1.43 (arogenate dehydrogenase) inhibitor, a nutraceutical, a micronutrient and a fundamental metabolite. It is an erythrose 4-phosphate/phosphoenolpyruvate family amino acid, a proteinogenic amino acid, a tyrosine and a L-alpha-amino acid. It derives from a L-tyrosinal. It is a conjugate base of a L-tyrosinium(1+). It is a conjugate acid of a L-tyrosinate(1-). It is an enantiomer of a D-tyrosine. It is a tautomer of a L-tyrosine zwitterion. | The molecule is an amino trisaccharide that is 2-acetamido-2-deoxy-D-glucopyranose in which the hydroxy groups at positions 3 and 4 have been converted into the corresponding beta-D-galactopyranosyl and alpha-L-fucopyranosyl derivatives, respectively. It is an amino trisaccharide and a member of acetamides. It derives from an alpha-L-Fucp-(1->4)-D-GlcpNAc. | The molecule is an amino trisaccharide consisting of N-acetylglucosamine having a fucosyl residue attached at the 4-position via a beta-linkage and a galactosyl residue attached at the 3-position via an alpha-linkage. It has a role as an epitope. It is an amino trisaccharide and a glucosamine oligosaccharide. | The molecule is an amino trisaccharide consisting of N-acetylglucosamine having a fucosyl residue attached at the 4-position via an alpha-linkage and a galactosyl residue attached at the 3-position via a beta-linkage. It has a role as an epitope and an antigen. It is an amino trisaccharide and a glucosamine oligosaccharide. |

Figure 7: Cases for the Mol2Cap task.

## C.4 CAP2MOL CASES

Figure 8: Cases for the Cap2Mol task.

# D PROMPT TEMPLATES

We list all the prompt templates applied in our work here. Figure 9 is the prompt template for Zero-shot Alignment Extraction, while Figure 10 shows the prompt templates for In-Context Selective Reflection. Additionally, Figure 11 shows the templates for MolReFlect without context examples, and Figure 12 illustrates the prompt templates for CoT-ICMT. All the templates are designed to fit the chat template of LLMs with roles including system, user, and assistant.

> \<System>: You are an assitant of a chemist user. Please follow the instruction of the chemist and complete the chemistry tasks.

> \<User>: Here is a molecule represented by SMILES strings:
> ```
> {molecule}
> ```
>
> Please help extract fine-grained alignments from the molecule structure. The fine-grained alignments should indicate the structure patterns, such as important functional groups, number of carbon atoms, configuration, group/family, derivatives, and anything that may affect the chemical features of the molecule. Your answer should follow the markdown format, using '*' to organize your answer into several points.

(a) Mol2Cap

> \<System>: You are an assitant of a chemist user. Please follow the instruction of the chemist and complete the chemistry tasks.

> \<User>: Here is a molecule caption that describes the properties of the molecule:
> ```
> {caption}
> ```
>
> Please help extract fine-grained alignments from the molecule caption. The fine-grained alignments should describe the structure and chemical features of the molecule. Your answer should follow the markdown format, using '*' to organize your answer into several points.

(b) Cap2Mol

Figure 9: Prompt templates for Zero-shot Alignment Extraction.

<System>: You are an assitant of a chemist user. Please follow the instruction of the chemist and complete the chemistry tasks.

<User>: Example {n}:
```
Molecule: {molecule_n}
Molecule fine-grained alignments: {alignments_n}
```

Based on above examples, now, here is a molecule represented by SMILES strings:
```
{molecule}
```
Please help extract fine-grained alignments from the molecule structure. The fine-grained alignments should indicate the structure patterns, such as important functional groups, number of carbon atoms, configuration, group/family, derivatives, and anything that may affect the chemical features of the molecule. You could gain insight from the similar examples, but the examples are not necessarily correct. You could first analyse the examples then give your final answer. Notably, your answer should follow the markdown format, using '*' to organize your answer into several points.

(a) Mol2Cap

<System>: You are an assitant of a chemist user. Please follow the instruction of the chemist and complete the chemistry tasks.

<User>: Example {n}:
```
Molecule Caption: {caption_n}
Caption fine-grained alignments: {alignments_n}
```

Based on above examples, now, here is a molecule caption that describes the properties of the molecule:
```
{caption}
```
Please help extract fine-grained alignments from the molecule caption. The fine-grained alignments should describe the structure and chemical features of the molecule. You could gain insight from the similar examples, but the examples are not necessarily correct. You could first analyse the examples then give your final answer. Notably, your answer should follow the markdown format, using '*' to organize your answer into several points.

(b) Cap2Mol

Figure 10: Prompt templates for In-Context Selective Reflection.

<System>: You are an assitant of a chemist user. Please follow the instruction of the chemist and complete the chemistry tasks.

<User>: Here is a molecule represented by SMILES strings:
```
{molecule}
```
Please help extract fine-grained alignments from the molecule structure. The fine-grained alignments should indicate the structure patterns, such as important functional groups, number of carbon atoms, configuration, group/family, derivatives, and anything that may affect the chemical features of the molecule.

<Assistant>: {alignments}

<User>: Now please generate the molecule caption. You answer should be concluded to the JSON format, such as {'caption': The molecule is 'MOLECULE CAPTION CONTENT'}.

(a) Mol2Cap

<System>: You are an assitant of a chemist user. Please follow the instruction of the chemist and complete the chemistry tasks.

<User>: Here is a molecule caption that describes the properties of the molecule:
```
{caption}
```
Please help extract fine-grained alignments from the molecule caption. The fine-grained alignments should describe the structure and chemical features of the molecule.

<Assistant>: {alignments}

<User>: Now please generate the SMILES representation of the molecule. You answer should be concluded to the JSON format, such as {'molecule': 'MOLECULE SMILES'}.

(b) Cap2Mol

Figure 11: Prompt templates for MolReFlect (w/o Fine-grained Alignments).

<System>: You are an assitant of a chemist user. Please follow the instruction of the chemist and complete the chemistry tasks.

<User>: Example {n}:
```

Molecule: {molecule_n}
Molecule fine-grained alignments: {alignments_n}
Molecule Caption: {caption_n}
```

Based on above examples, now, here is a molecule represented by SMILES strings:
```

{molecule}
```
Please help extract fine-grained alignments from the molecule structure. The fine-grained alignments should indicate the structure patterns, such as important functional groups, number of carbon atoms, configuration, group/family, derivatives, and anything that may affect the chemical features of the molecule.

<Assistant>: {alignments}

<User>: Now please generate the molecule caption. You could gain insight from the similar examples, but the examples are not necessarily correct.

(a) Mol2Cap

<System>: You are an assitant of a chemist user. Please follow the instruction of the chemist and complete the chemistry tasks.

<User>: Example {n}:
```

Molecule Caption: {caption_n}
Caption fine-grained alignments: {alignments_n}
Molecule: {molecule_n}
```

Based on above examples, now, here is a molecule caption that describes the properties of the molecule:
```

{caption}
```

Please help extract fine-grained alignments from the molecule caption. The fine-grained alignments should describe the structure and chemical features of the molecule.

<Assistant>: {alignments}

<User>: Now please generate the SMILES representation of the molecule. You could gain insight from the similar examples, but the examples are not necessarily correct.

(b) Cap2Mol

Figure 12: Prompt templates for Chain-of-Thought In-Context Molecule Tuning (CoT-ICMT).

