# OpenReview forum: "MolReFlect: Towards Fine-grained In-Context Alignment between Molecules and Texts"
_ICLR.cc/2025/Conference — ICLR 2025 Conference Withdrawn Submission_

### Official Review · Reviewer_8viw · 2024-10-29

**Soundness:** 2
**Presentation:** 2
**Contribution:** 2
**Rating:** 5
**Confidence:** 3

**Summary:**

The paper presents MolReFlect, a teacher-student framework that enhances the alignment between molecular structures and their captions, addressing challenges in molecule understanding and generation using Large Language Models (LLMs). By leveraging fine-grained alignments and innovative In-Context Selective Reflection technique, the approach achieves state-of-the-art performance on the ChEBI-20 dataset.

**Strengths:**

* The paper introduces MolReFlect teacher-student framework, a novel approach to molecular caption alignment, addressing a critical gap in existing methods that often overlook molecular substructures.

* The experimental evaluation against a limited baselines set showcases the effectiveness of the proposed approach for generative cross-modal tasks on molecules and texts.

* A thorough description of the proposed methodology with the detailed description of the experimental setting.

**Weaknesses:**

* The experimental evaluation of the proposed methodology in conducted on the single ChEBI-20 dataset and lacks comparison against more recent state-of-the-art molecule captioning/generation approaches.
* While SELFIES molecular string representations are mentioned in the paper's Introduction, the experiments are conducted on SMILES. This motivation for choosing SMILES is not explained.
* The proposed MolReFlect method is implemented using only a single set-up with Llama-3-70B-Instruct and Mistral-7B as teacher and student model, respectively. No experiments with larger student LLM/smaller teacher LLM are provided.
* High computational complexity and memory footprint of the proposed method. The implementation of the proposed pipeline has at least 77B parameters. For comparison, BioT5 is reported to have 252M parameters only.
* The proposed pipeline includes additional tasks, for instance, Named Entity Recognition for molecule substructure extraction and entity linking during In-context Selective Reflection. From the paper, the performance of LLMs on this indermideate tasks is not clear.

**Questions:**

* Add more experiments on other datasets, for instance, Mol-Instructions [5].
* Are the improvements over baseline models statistically significant?
* Add more experimental comparison against more chemical LMs: e.g., nach0 [1], Text+Chem T5 [2], SciFive [3], PRESTO [4], GitMol [6]. MoMu is mentioned in the Related Work but it is absent from experimental comparison.
* The usage of SELFIES is expected to give 100% SMILES validity due to the format specification. Is it possible to conduct additional experiments with SELFIES or explain the choice of SMILES in your experiments?
* Error Analysis/case study, revealing key challenges for the proposed framework/hard cases/interpretation of the cases where MolReFlect gains an improvement over baselines, would strengthen the paper.
* Line 220: the notation is not quite clear to me. What are alignments $K$ formally? A function? A tuple of pairs <text, molecule substructure>? A set?
* In section 4.2, more discussion on the adopted baselines would help: architecture, parameter count, hugging face checkpoint links, pretraining data, etc.

Typos:
* line 192: What is ICMA? The abbreviation is not introduced earlier.
* line 399: Mo2Cap -> Mol2Cap

References:

[1] Livne, Micha, et al. "nach0: Multimodal natural and chemical languages foundation model." Chemical Science 15.22 (2024): 8380-8389.

[2] Christofidellis, Dimitrios, et al. "Unifying molecular and textual representations via multi-task language modelling." International Conference on Machine Learning. PMLR, 2023.

[3] Phan, Long N., et al. "Scifive: a text-to-text transformer model for biomedical literature." arXiv preprint arXiv:2106.03598 (2021).

[4] Cao, He, et al. "PRESTO: Progressive Pretraining Enhances Synthetic Chemistry Outcomes." arXiv preprint arXiv:2406.13193 (2024).

[5] Yin Fang, Xiaozhuan Liang, Ningyu Zhang, Kangwei Liu, Rui Huang, Zhuo Chen, Xiaohui Fan, Huajun Chen:
Mol-Instructions: A Large-Scale Biomolecular Instruction Dataset for Large Language Models. ICLR 2024

[6] Liu, P., et al. "GIT-Mol: A multi-modal large language model for molecular science with graph, image, and text." Computers in Biology and Medicine 171 (2024): 108073-108073.

---

> ### Author Response · Authors · 2024-11-16
>
> Thank you for your insightful review. We appreciate your feedback and are pleased to address your questions point by point:
>
> 1. **Supplementary Dataset**:
>    - Thank you for your suggestion. We have incorporated another widely recognized dataset, `PubChem`, for our experiments, using the same backbone model, Mistral-7B-Instruct-v0.2. The results from this dataset similarly confirm the effectiveness of our method. The results are shown below:
>
> | Cap2mol                 | bleu  | EM    | Levenshtein | Maccs fts | rdk fts  | morgan fts | validity |
> |-------------------------|-------|-------|-------------|-----------|----------|------------|----------|
> | Mistral-7B              | 0.438 | 0.082   | 74.16       | 0.731     | 0.577    | 0.472      | 0.866     |
> | MolReFlect w/o CoT-ICMT | 0.744 | 0.145 | 30.23       | 0.799     | 0.662    | 0.560      | 0.955     |
> | MolReFlect              | 0.763 | 0.172 | 27.69       | 0.806      | 0.678    | 0.577      | 0.962     |
>
>
> | Mol2cap                 | bleu-2 | bleu-4 | rouge-1 | rouge-2 | rouge-l | meteor |
> |-------------------------|--------|--------|---------|---------|---------|--------|
> | Mistral-7B              | 0.361  | 0.288  | 0.471   | 0.325   | 0.419   | 0.421  |
> | MolReFlect w/o CoT-ICMT | 0.369   | 0.297  | 0.482    | 0.342   | 0.433    | 0.431  |
> | MolReFlect              | 0.414  | 0.343  | 0.511   | 0.374   | 0.458   | 0.470     |
>
>
> 2. **Performance Improvement**:
>    - Yes, our method has achieved an average of 8.9% improvement in the Cap2Mol task and 3.5% improvement over ICMA. We consider this enhancement to be significant. With more advanced Teacher LLMs, the performance improvement can only become larger.
>
> 3. **Applicability to Chemical LMs**:
>    - We appreciate your suggestion, but our method is not suitable for these chemical LMs. Our approach requires a more stringent context length, which is not compatible with models having a context length of less than 4096. Moreover, our focus is on the reasoning and in-context learning capabilities of LLMs, as emphasized in the ICMA. In this case, choosing general LLMs caters more to the inherent characteristics of MolReFlect.
> Lastly, we are sorry for not presenting MoMu for comparison, we will fix that in our revised version.
>
> 4. **Use of SELFIES**:
>    - We understand that SELFIES can produce more valid molecules. However, general LLMs are not suitable for SELFIES as they require a specially designed tokenizer, which would necessitate additional training for the LLM, contrary to our plug-and-play approach. Additionally, the generation of SELFIES by LLMs is not always 100% accurate, as seen in BioT5, where natural language text can sometimes be interspersed with molecular representations. Lastly, models have a better understanding of SMILES due to its widespread use, which increases the likelihood of learning SMILES-related knowledge from other corpora. We have also explained that 100% accurate molecules can be generated from SMILES using repeated generation and various decoding strategies.
>
> 5. **Case Study**:
>    - Thank you for your suggestion. Our case study is already quite detailed, with customized examples that clearly reflect key challenges such as the number of benzene rings and molecular properties. Our method effectively captures the details in the captions.
>
> 6. **Clarification on Variable 'K'**:
>    - We apologize for any confusion. In our paper, 'K' denotes a paragraph generated by the LLM, which contains the alignment from molecules to texts or text phrases to molecule substructures. You may also refer to Figure 5 in the Appendix to see examples of how this alignment looks.
>
> 7. **Parameter Efficiency**:
>    - We acknowledge that our method may not be the most parameter-efficient, but larger models can help solve questions better and the computation cost is reducing gradually. In response to your query, we have collected a table that addresses the issues you raised.
>
> | Model | architecture | parameter count | hugging face checkpoint links | pretraining data |
> |--|--|--|--|--|
> | MolT5-base | encoder-decoder | 248M | https://huggingface.co/laituan245/molt5-base | C4, Zinc |
> | BioT5-base | encoder-decoder | 252M | https://huggingface.co/QizhiPei/biot5-base | C4, Wrapped Text, PubChem |
> | Galactica-125M | decoder-only | 125M | https://huggingface.co/facebook/galactica-125m  | Galactica Corpus (Papers, Code, Reference Material Knowledge Bases, Filtered CommonCrawl, Prompts, Other) |
> | Mistral-7B-Instruct-v0.2 | decoder-only | 7.3B | https://huggingface.co/mistralai/Mistral-7B-Instruct-v0.2 | Not Specified |
> | llama3-8B-Instruct       | decoder-only    | 8B               | https://huggingface.co/meta-llama/Meta-Llama-3-8B         | Not Specified                                                                                             |
>
>
> We hope that our responses have adequately addressed your concerns and provided the necessary clarifications.

---

> > ### Comment · Reviewer_8viw · 2024-11-17
> > **Official Comment by Reviewer 8viw**
> >
> > Dear authors,
> >
> >
> >
> > Thank you for addressing my concerns. Let me clarify a few of concerns I got an incomplete response for.
> >
> > * **Applicability to Chemical LMs:** I meant adding all the listed chemical language models as baselines for comparison. An extension of baselines list would give more insights on the effectiveness of your proposed methodology.
> >
> >
> > * **Use of SELFIES:** Could you please describe why your model is not applicable for SELFIES? As far as I know, SELFIES is another string format, why can't we tokenize a SELFIES string with an existing tokenizer. Mol-Instructions (https://arxiv.org/pdf/2306.08018) adopted SELFIES. Here are the quotes from their paper: ``To circumvent these issues, we opt for SELFIES (Krenn et al., 2022) as our molecular descriptor.`` and ``To investigate whether Mol-Instructions can enhance LLM’s understanding of biomolecules, we perform instruction tuning on the three main domains of Mol-instructions, using LLama-7B (Touvron et al., 2023) as the foundation model. We additionally employ Alpaca-LoRA (Tloen, 2023), Baize-7B (Xu et al., 2023), ChatGLM-6B (Zeng et al., 2023), Vicuna (Chiang et al., 2023),``. So, it looks like the authors of Mol-Instructions managed to run general-purpose LLMs with SELFIES input, what is the key obstacle in the case of MolReFlect? If possible, support your claims regarding SELFIES with references to other papers.

---

> > > ### Author Response · Authors · 2024-11-18
> > >
> > > Thanks for your clarification. We appreciate every chance to respond to your concerns.
> > >
> > > #### **Adding Chemical LMs for baselines**:
> > > Thanks for your clarification and suggestions. We agree that the comparison with more baselines will definitely benefit this work. We will add these chemical LMs as baselines for comparison in our revised version.
> > >
> > > #### **Use of SELFIES**
> > > We want to clarify more about this question. We are not saying MolReFlect is not applicable for SELFIES, but `not suitable`.
> > >
> > > Actually, MolReFlect focuses on adopting general LLMs in the molecule-caption translation task, which requires the reasoning and in-context learning capability of LLMs. For general LLMs, they are not designed for SELFIES, but could see more SMILES strings in their pre-training corpus. Meanwhile, SMILES do not require a specifically designed tokenizer and the corresponding word embeddings.
> > >
> > > To be more specific, let us look at the SELFIES strings, where a carbon atom is represented as '[C]'. In contrast, a carbon atom is represented as 'C' in SMILES. As general LLM are not specifically designed for SELFIES, their tokenizers normally do not contain the token '[C]'. In this case, a single carbon atom in SELFIES '[C]' will cost 3 tokens. If you use SELFIES for general LLMs without modifying their tokenizers and embeddings,
> > > * **for the generation of molecules**,  it will increase the possibility of generating an invalid molecule, as the LLM needs to generate all three tokens of '[C]' correctly. Here is a simple example: general LLMs could generate SELFIES strings like '[C[C]'.
> > > * **for taking molecules as inputs**, it will increase the context length dramatically. *Please note that we are not saying that we can not take SELFIES as the input.* The question is why use SELFIES 1) when general LLMs are not familiar with them; 2) when SELFIES cost more input tokens than SMILES.
> > >
> > > However, we are also not saying that we can not modify the tokenizer of general LLMs by adding SELFIES as special tokens to make them applicable to SELFIES, but any modification to the existing tokenizer will actually require continual pre-train to let LLMs learn the newly added embeddings. This will pose two questions:
> > > 1. The continual pre-train will modify the parameters of the general LLMs, which will cause **catastrophic forgetting** of the previously learned knowledge [1] and can harm the original reasoning and in-context learning capability.
> > > 2. The extended embedding will add extra computation costs to the decoding process, as the LLM is required to assign each word in the vocabulary a probability.
> > >
> > > `Given the above reasons, we see SELFIES is not suitable for current general LLMs. If they support SELFIES seamlessly in the future, we are happy to use SELFIES. Anyway, the representation of molecules is not a key focus of this work.`
> > >
> > > If you are still interested in comparing SMILES and SELFIES, I would say that it is actually an interesting topic to research.
> > > Currently, chemical LLMs using SELFIES like BioT5 [2] and BioT5+ [3] require pre-training from scratch on SELFIES corpora to ensure the quality of embeddings. However, they do not ablate the usage of SELFIES. There are several works worth noticing:
> > > * In [4], they actually compare SMILES and SELFIES as LLM inputs. You may see that the performance of using SMILES is even better than SELFIES from their Table 7 and Table 8 in molecule property prediction tasks.
> > > * In [5], they find that "Invalid SMILES are a shortcoming of chemical language models and reframe them as a feature, not a bug".
> > >
> > > We sincerely hope you can read this explanation carefully and hope our response can resolve your concerns about this issue. Thanks again for your valuable input!
> > >
> > > #### **References**
> > > [1] Luo, Y., Yang, Z., Meng, F., Li, Y., Zhou, J., & Zhang, Y. (2023). An empirical study of catastrophic forgetting in large language models during continual fine-tuning. arXiv preprint arXiv:2308.08747.
> > >
> > > [2] Pei, Q., Zhang, W., Zhu, J., Wu, K., Gao, K., Wu, L., ... & Yan, R. (2023). Biot5: Enriching cross-modal integration in biology with chemical knowledge and natural language associations. arXiv preprint arXiv:2310.07276.
> > >
> > > [3] Pei, Q., Wu, L., Gao, K., Liang, X., Fang, Y., Zhu, J., ... & Yan, R. (2024). Biot5+: Towards generalized biological understanding with iupac integration and multi-task tuning. arXiv preprint arXiv:2402.17810.
> > >
> > > [4] Leon, M., Perezhohin, Y., Peres, F., Popovič, A., & Castelli, M. (2024). Comparing SMILES and SELFIES tokenization for enhanced chemical language modeling. Scientific Reports, 14.
> > >
> > > [5] Skinnider, M. A. (2024). Invalid SMILES are beneficial rather than detrimental to chemical language models. Nature Machine Intelligence, 6(4), 437-448.

---

### Official Review · Reviewer_hpY1 · 2024-10-31

**Soundness:** 2
**Presentation:** 3
**Contribution:** 2
**Rating:** 5
**Confidence:** 3

**Summary:**

The paper intends to enable fine-grained alignments and introduce a large teacher LLM to label the detailed alignments. Then the paper proposes in-context selective reflection, to let a student LLM select from in-context reflection and previous extraction results. Finally, the paper introduces Chain-of-Thought, which also increases the performance.

**Strengths:**

1. The performance of generation is impressive.

2. The paper is well-motivated to enable the fine-grained alignment.

**Weaknesses:**

1. The main concern remains in potential data leakage. The paper widely uses the pretrained large model, such as teacher model (Llama-3-70B), Mole-BERT and student model (Mistral-7B). The statement of data leakage in the paper is not convincing enough. Even though the teacher model cannot perform well in generation, given that the teacher model can do zero-shot alignment labeling, there's still possible potential leakage. Additionally, the pre-trained Mole-BERT may also introduces leakage, which should be further discussed. Although there may leakage in student model, the authors have already presented the effectiveness of the proposed pipeline. However, the comparison of the baselines may not be fair.

2. In Line 265, why an unsupervised metric can avoid leakage from Galactiva-125M?

3. Is this paper possible also evaluated on retrieval tasks?

4. The justification in related work about 2D and 3D graph is not fair. Note that the molecular data is limited compared with image-text pairs, and the structural information is important for molecules.

I am happy to increase the score if the problems are solved.

**Questions:**

See weaknesses.

---

> ### Author Response · Authors · 2024-11-16
>
> Thank you for your thoughtful review. We appreciate the opportunity to address your concerns and clarify our findings. Below are our point-by-point responses:
>
> ### Response to Reviewer Comments
>
> 1. **Regarding Data Leakage**:
>    - We share your concern about data leakage and agree that it is unacceptable in experimental settings. However, we disagree with the comment that data leakage may have occurred in our LLM experiments. In fact, we have made our best efforts to reduce the possibility of data leakage.
>      - Firstly, we chose the mistral-7B and llama3 series of LLMs to minimize the risk of data leakage. Actually, the pre-training of the mistral-7B and llama-3 series, while not fully disclosed, are known to be relatively clean, which means that they did not use the ChEBI-20 dataset for pre-training. Meanwhile, the cross-validation between Mistral-7B and Llama3-8B also shows the model agnosticism of MolReFlect, which means that the performance improvement is not from data leakage.
>      - Secondly, our zero-shot experimental setup is indeed aggressive. If the model had been trained on the ChEBI-20 dataset, it should have produced related training data and achieved better performance under the zero-shot setting. However, the poor zero-shot performance indicates that no explicit data leakage occurred in the LLM.
>      - Lastly, the generation of correct molecules by the above-mentioned LLM does not necessarily imply data leakage. MolReGPT has shown that GPT-4 can generate correct molecules through reasoning, which is within their capabilities. However, this does not equate to test data leakage, as they could have generalised to this task due to the massive pre-training.
>      - We believe the existing experimental evidence is sufficient, and we request that our efforts not be dismissed based on speculation alone.
>      - Regarding Mole-BERT, we can confidently state that no test set molecules were retrieved in the retrieval process, and Mole-BERT has never been trained on ChEBI-20. Any concerns about test data leakage with this model are unfounded. Should you have evidence to the contrary, we welcome your feedback.
>
> 2. **Regarding Unsupervised Metrics and Galactica-125M**:
>    - We acknowledge the importance of ensuring no ground truth is introduced during the selective process. Therefore, we used perplexity as an information-theoretic metric, which does not involve ground truth and thus prevents data leakage.
>    - Concerning the potential data leakage in Galactica-125M, we believe this concern is unwarranted. The training data for this model is fully public, and `ChEBI-20 is not included`. We have conducted similar attacks on this model's training data and found no evidence of test data leakage. If you have evidence to suggest otherwise, we invite you to share it with us. Furthermore, previous works like MolCA and 3D-MoLM have also used this backbone without issues.
>
> 3. **Regarding Retrieval Tasks**:
>    - We appreciate your suggestion, but it is not feasible to validate our approach on retrieval tasks. Our focus is solely on the molecule-caption translation task, and we do not aim to enhance the embedding representations of molecules or captions. The context input consists of a series of examples and the current molecule, and summarizing these into embeddings for retrieval is not meaningful.
>    - Additionally, our method employs a decoder-only model, which has inherent disadvantages in embedding compared to encoder-only or encoder-decoder models.
>
> 4. **Regarding Unfair comparison**:
>    - We acknowledge the importance of structural information and our use of a graph-encoder for retrieval. However, we contend that existing work does not fully utilize 2D and 3D information.
>      - SMILES and graph representations are losslessly convertible, and LLMs are designed to process sequential information. The alignment dataset required for LLMs to understand 2D graphs is not sufficient, as you also mentioned, and the alignment process could lead to knowledge loss in LLMs.
>      - Similarly, issues apply to 3D information. Current methods use RDKit-estimated 3D information, which can be significantly different from ground truth, raising questions about accuracy.
>
> We thank you again for your valuable comments and hope that our responses have adequately addressed your concerns. Should you have any further questions, we are always open for discussion.

---

> > ### Comment · Reviewer_hpY1 · 2024-11-20
> > **Thanks for your response**
> >
> > Firstly, I would like to thank the authors for their response.
> >
> > I am surprised to find that the authors said, "we request that our efforts not be dismissed based on speculation alone." I want to clarify that in my initial review, I have already acknowledged the efforts made by the authors. I stated, "the authors have already presented the effectiveness of the proposed pipeline." However, the data leakage problem is very common in LLMs. Therefore, it is important to ensure a fair evaluation, which is also a significant challenge in NLP. The authors should at least discuss how they prevent or alleviate data leakage in their manuscript. Unfortunately, I did not find any solid explanation.
> >
> > In the rebuttal, the authors said, "Actually, the pre-training of the mistral-7B and llama-3 series, while not fully disclosed, are known to be relatively clean, which means that they did not use the ChEBI-20 dataset for pre-training." I am not sure why the authors claim this. Potential data leakage does not necessarily mean that Llama3 was directly trained on the ChEBI-20 dataset. As we know, LLMs are trained on a large amount of scientific papers, which may mean they have access to abundant knowledge.
> >
> > If the authors still believe the test data is not leaked, I strongly encourage you to conduct an extremely simple experiment. First, ask Llama3, "Do you know the ChEBI-20 dataset?" Then ask, "Could you please give me some examples of test data in the ChEBI-20 dataset?" Llama3 indicates that it is familiar with the ChEBI-20 dataset and provides some example molecules with their properties.
> >
> > Again, I want to emphasize that the data leakage problem is extremely important and should be carefully considered when using a powerful LLM. The authors should dive into this problem and provide convincing explanations. I will decide whether to increase, decrease, or maintain the scores based on further feedback from the authors.

---

> ### Author Response · Authors · 2024-11-20
>
> Thanks for your insightful comments and your clarification. We can't agree more with you regarding the data leakage problem.
>
> **Firstly**, we want to clarify what the "efforts" mean. We have made them clearly in our rebuttal that we have conducted zero-shot experiments (namely Direct Prompting in Table 5 and 6) on the ChEBI-20 dataset. If, as you claimed, the ChEBI-20 dataset is leaked during the training process of llama3-70B, the zero-shot performance should be rather high, because they have already seen these test samples during the training process. However, what we see is the performance is quite poor. Only 3% molecule is generated correctly and the BLEU-2 socre in the Mol2Cap task is even close to zero. By showing this, we want to emphasize that **we have already made efforts to ensure that the models we use are not trained on the ChEBI-20 dataset before and they did not see the test examples elsewhere**.
>
> **Secondly**, we are willing to add a simple test experiment as you mentioned. Here is our conversation with Llama-3-70B-Instruct:
> ```
> We: Do you know the ChEBI-20 dataset?
>
> Llama-3-70B: A question from the realm of bioinformatics and cheminformatics!
> Yes, I'm familiar with the ChEBI-20 dataset. ChEBI (Chemical Entities of Biological Interest) is a freely available dictionary of molecular entities focused on 'small' chemical compounds. The ChEBI-20 dataset is a benchmarking dataset derived from the ChEBI database, specifically designed for evaluating the performance of machine learning models in predicting chemical properties and bioactivities.
> The ChEBI-20 dataset consists of 20 classification tasks, each corresponding to a specific biological or chemical property, such as:
> Is the compound a neurotransmitter?
> Is the compound a hormone?
> Is the compound a antibiotic?
> ...and so on.
> Each task is a binary classification problem, where the goal is to predict whether a given molecule possesses a particular property or not. The dataset provides a set of molecular structures, represented as SMILES (Simplified Molecular Input Line Entry System) strings, along with their corresponding labels (positive or negative) for each of the 20 tasks.
> The ChEBI-20 dataset is widely used in the cheminformatics and bioinformatics communities to evaluate the performance of machine learning models, such as graph neural networks, random forests, and support vector machines, on tasks related to chemical property prediction and bioactivity analysis.
> Do you have a specific question about the ChEBI-20 dataset or are you working on a project related to this dataset?
>
> We: Could you please give me some examples of test data in the ChEBI-20 dataset?
>
> Llama-3-70B: The ChEBI-20 dataset is a collection of molecular structures, represented as SMILES strings, along with their corresponding labels for each of the 20 classification tasks. Here are a few examples of test data in the ChEBI-20 dataset:
> Task 1: Is the compound a neurotransmitter?
> SMILES: CC(=O)Nc1ccc(cc1)S(=O)(=O)N (Label: 1, meaning it is a neurotransmitter)
> SMILES: CC(=O)OC1=CC=CC=C1 (Label: 0, meaning it is not a neurotransmitter)
> Task 5: Is the compound a hormone?
> SMILES: CC(=O)Nc1ccc(cc1)S(=O)(=O)N (Label: 1, meaning it is a hormone)
> SMILES: COC(=O)c1ccc(cc1)N (Label: 0, meaning it is not a hormone)
> Task 10: Is the compound a antibiotic?
> SMILES: CC(=O)Nc1ccc(cc1)S(=O)(=O)N (Label: 0, meaning it is not an antibiotic)
> SMILES: CC(=O)Nc1c(=O)[nH]c(=O)c1 (Label: 1, meaning it is an antibiotic)
> Task 15: Is the compound a flavonoid?
> SMILES: C1=CC(=C(C(=C1)O)O)O (Label: 1, meaning it is a flavonoid)
> SMILES: CC(=O)Nc1ccc(cc1)S(=O)(=O)N (Label: 0, meaning it is not a flavonoid)
> These are just a few examples of the test data in the ChEBI-20 dataset. The actual dataset contains thousands of molecular structures, each with its corresponding labels for the 20 classification tasks.
> Keep in mind that the SMILES strings are just a representation of the molecular structure, and the actual molecules can have different conformations, tautomers, or stereoisomers. The labels are based on the biological or chemical properties of the molecules, which can be obtained from various sources, including literature, databases, or experimental data.
> ```
> You may see that all of the responses genertaed by Llama-3-70B-Instruct are actually hallucination. ChEBI-20 is not a classification dataset. The samples generated are not found in the ChEBI-20 dataset.
>
> We want to clarify again that our zero-shot (i.e., Direct Prompting) is actually more aggressive than this simple test experiment. If really, the test samples are leaked, the zero-shot performance of Llama-3-70B-Instruct shouldn't be that low.
>
> We sincerely hope that our responses could resolve your concerns.

---

> ### Author Response · Authors · 2024-11-20
>
> Our conversation with Llama-3-8B-Instruct
> ```
> We: Do you know the ChEBI-20 dataset?
>
> Llama-3-8B: A very specific question!
> Yes, I'm familiar with the ChEBI-20 dataset. ChEBI-20 is a dataset of 20,000 small molecules, which is a subset of the Chemical Entities of Biological Interest (ChEBI) database. ChEBI is a freely available dictionary of "small molecular entities" that are of biological interest, including metabolites, natural products, and other small molecules.
> The ChEBI-20 dataset was created by the European Bioinformatics Institute (EMBL-EBI) and is widely used in bioinformatics, cheminformatics, and machine learning research. The dataset contains a diverse set of small molecules, including metabolites, natural products, and synthetic compounds, with a focus on molecules that are relevant to biological systems.
> The ChEBI-20 dataset is often used as a benchmark for evaluating the performance of machine learning models, such as neural networks, for tasks like molecule property prediction, similarity search, and virtual screening. It's also used in research on molecular structure representation, molecular similarity, and molecular property prediction.
> Do you have a specific question about the ChEBI-20 dataset or its applications?
>
> We: Could you please give me some examples of test data in the ChEBI-20 dataset
>
> Llama-3-8B: The ChEBI-20 dataset contains a diverse set of 20,000 small molecules, each represented as a SMILES string (Simplified Molecular Input Line Entry System). Here are a few examples of molecules in the ChEBI-20 dataset:
> Glucose: OC[C@H]1O[C@H](C(O)=O)[C@H](O)[C@H]1O
> Aspirin: CC(=O)Oc1ccc(cc1)C(=O)O
> Adenosine: C10=NC13=C2N1C=NC2=NC3=C(N=C3N)N
> L-Aspartic acid: NC(=O)C(C(=O)O)C(O)=O
> Cholesterol: `CC(C)C1CCC2C1(CCC3C2CCC4C3(CCC5C4(CCC6C5(CCC7C6(CCC8C7(CCC9C8(CCC10C9)C)C)C)C)C)C)C)C)C)C)C)C)C)C)C)C)C)C)C)C)C)C)C)C)C)C)C)C)C)C)C)C)C)C)C)C)C)C)C)C)C)C)C)C)C)C)C)C)C)C)C)C)C)C)C)C)C)C)C)C)C)C)C)C)C)C)C)C)C)C)C)C)C)C)C)C)C)C)C)C)C)C)C)C)C)C)C)C)C)C)C)C)C)C)C)C)C)C)C)C)C)C)C)C)C)C)C)C)C)C)C)C)C)C)C)C)C)C)C)C)C)C)C)C)C)C)C)C)C)C)C)C)C)C)C)C)C)C)C)C)C)C)C)C)C)C)C)C)C)C)C)C)C)C)C)C)C)C)C)C)C)C)C)C)C)C)C)C)C)C)C)C)C)C)C)C)C)C)C)C)C)C)C)C)C)C)C)C)C)C)C)C)C)C)C)C)C)C)C)C)C)C)C)C)C)C)C)C)C)C)C)C)C)C)C)C)C)C)C)C)C)C)C)C)C)C)C)C)C)C)C)C)C)C)C)C)C)C)C)C)C)C)C)C)C)C)C)C)C)C)C)C)C)C)C)C)C)C)C)C)C)C)C)C)C)C)C)C)C)C)C)C)C)C)C)C)C)C)C)C)C)C)C)C)C)C)C)C)C)C)C)C)C)C)C)C)C)C)C)C)C)C)C)C)C)C)C)C)C)C)C)C)C)C)C)C)C)C)C)C)C)C)C)C)C)C)C)C)C)C)C)C)C)C)C)C)C)C)C)C)C)C)C)C)C)C)C)C)C)C)C)C)C)C)C)C)C)C)C)C)C)C)C)C)C)C)C)C)C)C)C)C)C)C)C)C)C)C)C)C)C)C)C)C)C)C)C)C)C)C)C)C)C)C)C)C)C)C)C)C)C)C)C)C)C)C)C)C)C)C)C)C)C)C)C)C)C)C)C)C)C)C)C)C)C)C)C)C)C)C)C)C)C)C)C)C)C)C)C)C)C)C)C)C)C)C)C)C)C)C)C)C)C)C)C)C)C)C)C)C)C)C)C)C)C)C)C)C)C)C)C)C)C)C)C)C)C)C)C)C)C)C)C)C)C)C)C)C)C)C)C)C)C)C)C)C)C)C)C)C)C)C)C)C)C)C)C)C)C)C)C)C)C)C)C)C)C)C)C)C)C)C)C)C)C)C)C)C)C)C)C)C)C)C)C)C)C)C)C)C)C)C)C)C)C)C)C)C)C)C)C)C)C)C)C)C)C)C)C)C)C)C)C)C)C)C)C)C)C)C)C)C)C)C)C)C)C)C)C)C)C)C)C)C)C)C)C)C)C)C)C)C)C)C)C)C)C)C)C)C)C)C)C)C)C)C)C)C)C)C)C)C)C)C)C)C)C)C)C)C)C)C)C)C)C)C)C)C)C)C)C)C)C)C)C)C)C)C)C)C)C)C)C)C)C)C)C)C)C)C)C)C)C)C)C)C)C)C)C)C)C)C)C)C)C)C)C)C)C)C)C)C)C)C)C)C)C)C)C)C)C)C)C)C)C)C)C)C)C)C)C)C)C)C)C)C)C)C)C)C)C)C)C)C)C)C)C)C)C)C)C)C)C)C)C)C)C)C)C)C)C)C)C)C)C)C)C)C)C)C)C)C)C)C)C)C)C)C)C)C)C)C)C)C)C)C)C)C)C)C)C)C)C)C)C)C)C)C)C)C)C)C)C)C)C)C)C)C)C)C)C)C)C)C)C)C)C)C)C)C)C)C)C)C)C)C)C)C)C)C)C)C)C)C)C)C)C)C)C)C)C)C)C)C)C)C)C)C)C)C)C)C)C)C)C)C)C)C)C)C)C)C)C)C)C)C)C)C)C)C)C)C)C)C)C)C)C)C)C)C)C)C)C)C)C)C)C)C)C)C)C)C)C)C)C)C)C)C)C)C)C)C)C)C)C)C)C)C)C)C)C)C)C)C)C)C)C)C)C)C)C)C)C)C)C)C)C)C)C)C)C)C)C)C)C)C)C)C)C)C)C)C)C)C)C)C)C)C)C)C)C)C)C)C)C)C)C)C)C)C)C)C)C)C)C)C)C)C)C)C)C)C)C)C)C)C)C)C)C)C)C)C)C)C)C)C)C)C)C)C)C)C)C)C)C)C)C)C)C)C)C)C)C)C)C)C)C)C)C)C)C)C)C)C)C)C)C)C)C)C)C)C)C)C)C)C)C)C)C)C)C)C)C)C)C)C)C)C)C)C)C)C)C)C)C)C)C)C)C)C)C)C)C)C)C)C)C)C)C)C)C)C)C)C)C)C)C)C)C)C)C)C)C)C)C)C)C)C)C)C)C)C)C)C)C)C)C)C)C)C)C)C)C)C)C)C)C)C)C)C)C)C)C)C)C)C)C)C)C)C)C)C)C)C)C)C)C)C)C)C)C)C)C)C)C)C)C)C)C)C)C)C)C)C)C)C)C)C)C)C)C)C)C)C)C)C)C)C)C)C)C)C)C)C)C)C)C)C)C)C)C)C)C)C)C)C)C
> ```
>
> You may see that the details are also not matched. The generated samples are mostly invalid molecules and they can not be found in the ChEBI-20 dataset.

---

> ### Author Response · Authors · 2024-11-20
>
> Our conversation with Mistral-7B-Instrcut-v0.2
> ```
> We: Do you know the ChEBI-20 dataset?
>
> Mistral-7B: Yes, I'm familiar with the ChEBI-20 dataset. The Chemical Entities of Biological Interest - 20 (ChEBI-20) dataset is a subset of the ChEBI (Chemical Entities of Biological Interest) database, which is a manually curated database of small chemical entities that are of interest in the field of biology and medicine. The ChEBI-20 dataset specifically consists of 20,000 small molecules that have been selected based on their relevance to human health and disease. It is commonly used in the field of cheminformatics and computational chemistry for various applications such as drug discovery, toxicity prediction, and biomarker identification.
>
> We: Could you please give me some examples of test data in the ChEBI-20 dataset?
>
> Mistral-7B: The ChEBI-20 dataset is a subset of the Chemical Entities of Biological Interest (ChEBI) database, which is a manually curated database of small chemical structures that are of biological interest. The ChEBI-20 dataset specifically refers to a subset of 20,000 small molecules that were selected from ChEBI for use in benchmarking and evaluating the performance of computational methods for structure-activity relationship (SAR) analysis and property prediction.
> Unfortunately, I cannot provide specific examples of test data from the ChEBI-20 dataset as it is not publicly available for download due to its size and the need to protect the intellectual property of the data. However, you can access the dataset through the European Bioinformatics Institute (EBI) by following the instructions provided on their website:
> https://www.ebi.ac.uk/chebi/chebi-20/
> The website provides information on how to obtain access to the dataset, which includes filling out a form and agreeing to certain terms and conditions. Once you have obtained access, you will be able to download the dataset and use it for your research purposes.
> The dataset includes various types of data for each molecule, including its ChEBI identifier, SMILES string, InChIKey, molecular formula, molecular weight, and various calculated properties such as logP, topological polar surface area, and molecular descriptors. It also includes experimental data such as IC50 values and other biological activity data for some of the molecules.
> ```
>
> You may see that the details are also not matched and the url it generated also does not exist.
>
> **By providing the above conversation examples, we want to relieve your concern on the potential data leakage. We fully understand and appreciate your reviews and your contributions to the community. We are also aware that we are actually at a bad situation. No matter what decision you will make, we will fully respect it.**

---

### Official Review · Reviewer_XB7S · 2024-11-03

**Soundness:** 2
**Presentation:** 3
**Contribution:** 2
**Rating:** 3
**Confidence:** 4

**Summary:**

This paper presents MolReFlect, a framework that refines molecule-caption alignments through a teacher-student model. It introduces an approach of zero-shot alignment extraction, in-context refinement, and Chain-of-Thought In-Context Molecule Tuning (CoT-ICMT), claiming state-of-the-art results on the ChEBI-20 dataset.

**Strengths:**

1. The proposed methods work reasonably well.
2. The paper contains plenty of ablations to validate the effectiveness of different components.

**Weaknesses:**

1. While the model performs well on generation tasks, its effectiveness on other molecular tasks, such as molecule retrieval and property prediction tasks, is not evaluated.

2. The model is only evaluated on the ChEBI-20 dataset, making it difficult to fully assess the model's generative capabilities.

3. For generation tasks, language models tend to memorize the training dataset. Presenting the scores of molecule novelty would be more comprehensive and show that MolReFlect can perform well in generation tasks.

4. Lack of human evaluation: For tasks like molecule captioning, incorporating human expert evaluation could provide additional validation of the model's performance and practical utility.

5. Training code and implementation for reproducing the paper are missing.

6. The training requires expensive computations. The required computational comparisons of the proposed method and other methods should be presented.

**Questions:**

1. In Section 3.2 on molecule retrieval, you use a pre-trained Mole-BERT as the graph encoder to calculate the cosine similarities between molecule graph embeddings. Why is the graph encoder employed here when the SMILES string is taken as input, as stated in the abstract (Line 23)?
1. How does MolReFlect perform on more complex molecular structures or rare chemical compounds not well-represented in the training data?
2. How does the performance of MolReFlect change when dealing with highly technical or domain-specific textual descriptions that might require expert knowledge to interpret?
3. Have you explored the potential of MolReFlect for other cross-modal (text and corresponding data modalities) tasks beyond the molecular domain? If so, what were the results or challenges encountered?

---

> ### Author Response · Authors · 2024-11-16
>
> We are grateful for your constructive feedback. Below, we address each of your points and provide clarifications and additional information.
>
> ### Response to Weaknesses
>
> 1. **W1**: We would like to reiterate that our primary objective is to bridge the gap between molecules and natural language texts, rather than to assert superiority in molecule property prediction tasks. While we acknowledge the potential of exploring the performance of MolReFlect in other tasks, the molecular property prediction task actually requires the prediction of probabilities, which is naturally not an advantage of LLMs, and in fact, we can not get the exact probability from merely comparing two tokens (i.e., "Yes" and "No"). However, we are happy to share some preliminary results with you. We will get back to this comment once the experiments are settled.
> 2. **W2**: We appreciate your suggestion regarding molecule novelty. However, for the Cap2Mol task, which is a targeted generation task, the goal is to generate the given molecule from the caption. Since all molecules in the ChEBI-20 dataset are known, the concept of novelty is not applicable to this task.
> 3. **W3**: We agree that incorporating human evaluation could benefit the work. However, due to the complexity and cost associated with chemical expertise, we have relied on established evaluation frameworks for molecule-caption translation. Since there have been many works that follow the same evaluation process on this topic, we believe the metrics used in this work are sufficient to evaluate the performance of the generation.
> 4. **W4**: We apologize for any confusion. The code was intended for release upon publication. In response to your concern, we have already collected and submitted the required code via the anonymous GitHub for your reference: https://anonymous.4open.science/r/MolReFlect-BE83
> 5. **Regarding W5**: It is important to note that the computational cost cannot be solely attributed to the number of model parameters. The domain continual pretraining and modality alignment require larger external datasets, which can be more computationally intensive. In contrast, MolReFlect primarily involves a fine-tuning process.
> ### Responses to Questions
> 6. **Q1**: We address this question in three parts. Firstly, the conversion between molecular SMILES and molecular graphs is lossless, representing two representations of the same molecule. Thus, we can use molecular graphs for retrieval despite LLMs processing SMILES input. Secondly, molecular graphs inherently contain more information about atomic connections, making them more efficient for model understanding and structural similarity, which is crucial for molecular properties. Lastly, for more comparisons, please refer to the work ICMA, whose settings we followed.
> 7. **Q2**: To address this question, we extracted molecules with a length exceeding 100 characters from the test set, which forms a subset of 747 molecules. Even for these complex molecules, we achieved the best results. This is due to the combined effect of reflection and context examples.
>
> | Mol2cap     | bleu-2 | bleu-4 | rouge-1 | rouge-2 | rouge-l | meteor |
> |-------------|--------|--------|---------|---------|---------|--------|
> | MolT5-large | 68.37  | 62.4   | 71.49   | 59.03   | 66.09   | 68.51  |
> | GPT-4-0314  | 66.99  | 60.33  | 69.2    | 55.51   | 63.02   | 67.27  |
> | ICMA        | 68.68  | 63.24  | 72.7    | 61.07   | 67.32   | 71.33  |
> | MolReFlect  | `72.66`  | `67.5`   | `74.99`  | `64.17`   | `69.95`   | `73.97`  |
>
> 8. **Q3**: We must emphasize that all captions used in our study are domain-specific and highly technical, which is essential for the task at hand. However, we followed a similar setting to molecules and extracted captions with a length exceeding 500 characters from the test set, which forms a subset of 221 captions, where MolReFlect also achieves a better result.
>
> | Cap2mol     | bleu  | EM    | Levenshtein | Maccs fts | rdk fts  | morgan fts | validity |
> |-------------|-------|-------|-------------|-----------|----------|------------|----------|
> | MolT5-large | 62.38 | 11.31 | 50.71       | 80.13     | 66.88    | 58.35      | 79.19    |
> | GPT-4-0314  | 79.35 | 12.22 | 40          | 85.81     | 70.44    | 63         | 76.92    |
> | ICMA        | 68.41 | 23.53 | 70.79       | 88.32     | 73.61    | 66.98      | 87.33    |
> | MolReFlect  | `85.78` | `31.22` | `26.14`       | `89.21`     | `76.73`    | `71.21`      | `92.31`    |
>
> 9. **Q4**: This is an intriguing question. While it is true that a similar framework could be used with VLMs to summarize and highlight key aspects of modalities such as images to enhance image captioning tasks, we believe this extends beyond the scope of our current work. We would like to reiterate that MolReFlect is focused on molecular tasks. Should you be interested in this direction, we encourage and welcome similar explorations.

---

> > ### Author Response · Authors · 2024-11-17
> >
> > Here are some of the molecule property prediction results of MolReFlect on the MoleculeNet dataset. Here, we applied Mistral-7B as the backbone model. We compared MolReFlect with ICMA and the original Mistral-7B on the BACE and BBBP tasks. The results are shown in the Table below:
> >
> > | MoleculeNet | BACE   | BBBP   |
> > |-------------|--------|--------|
> > | Mistral7B   | 0.4926 | 0.4829 |
> > | ICMA        | 0.7995 | 0.6775 |
> > | MolReFlect  | `0.8795` | `0.8925` |
> >
> > It is obvious to see that MolReFlect also shows its potential in molecular property prediction tasks. We hope these results can resolve the W1 and W2 raised in your comments. Thanks again for your valuable suggestions!

---

### Official Review · Reviewer_zyE2 · 2024-11-03

**Soundness:** 3
**Presentation:** 3
**Contribution:** 2
**Rating:** 6
**Confidence:** 4

**Summary:**

The authors present MolReFlect, a novel teacher-student framework designed to refine the in-context alignments between molecular substructures and their corresponding textual descriptions. They show that within this framework the training of a small student network may benefit from the alignments, generated by a teacher model. The model trained with this framework shows a better performance on Mol2Cap and Cap2Mol tasks on the ChEBI-20 dataset compared to the baseline trained with SFT.

**Strengths:**

A new framework for training Chemical LLMs, that allows to successfully  finetune relatively small models with limited resources.
Good results for the molecular captioning tasks without domain pretraining.

**Weaknesses:**

All the baselines except ICMA differ from MolReFlect by at least an order of magnitude by the number of parameters.
The ablation studies are partially unconvincing: they are done only for one setup Cap2Mol or Mol2Cap
The results for w/o In-Context Reflection and w/o Selection setups are practically indistinguishable from the full model, especially in the table 4.

**Questions:**

1. The idea of using the retrieval in  Chain-of-Thought In-Context Molecule Tuning seems to be similar to the RAG. It would be interesting to compare these frameworks.
2. The authors don’t site arxiv.org/abs/2408.11866 where the better performance is achieved with a similar framework, but probably lager models.
3. Please, fix the bold numbers in Table 4 for the Validity metric.
4. It is interesting to check how robust the trained model is. (i.e. in terms of Ganeeva, Veronika, et al. "Chemical Language Models Have Problems with Chemistry: A Case Study on Molecule Captioning Task." The Second Tiny Papers Track at ICLR 2024).

---

> ### Author Response · Authors · 2024-11-16
>
> Thank you for your valuable comments. We appreciate the opportunity to clarify the weaknesses and address your insightful questions.
> ### Response to Weaknesses
>
> #### W1 - Model Size Concerns
> Firstly, regarding W1, we acknowledge that MolReFlect employs larger backbone models compared to previous baselines. Due to the unique nature of our method, which relies on the reasoning and in-context learning capabilities of LLMs, the models suitable for our approach are rather limited. Mistral-7B and llama3-8B are the most appropriate under our budget. We do not intentionally compare our method with previous baselines, as the difference in backbone models makes it challenging to create a completely fair control experiment environment. Returning to our motivation, we aim to better apply LLMs in the molecular field, particularly in the task of molecule-caption translation.
>
> #### W2 - Effectiveness of the Approach
> As for W2, we contend that although the experimental effect gap is not significant, this outcome is anticipated. On one hand, the llama3-70B model is not well-versed in molecular tasks and often refuses to respond, lacking molecular common sense. Consequently, the keypoints we extracted undergo minimal changes during the unsupervised self-refine process. Meanwhile, the presence of context examples also ensures a stable refinement process. On the other hand, since we select the better combination between the two generations, a modest improvement is expected. However, it is evident that the results after selection are always superior to those without, indicating a stable trend and providing a higher information gain.
>
> ### Responses to Questions
> 1. **Comparison with Other RAG Methods:**
> We agree with your suggestion to include comparisons with other RAG methods. In fact, MolReGPT and ICMA are also RAG methods, and we have compared our approach with both frameworks. Our method focuses on not introducing explicit external knowledge (such as knowledge graphs or search engines) but instead relies on the dataset and the reasoning capabilities of LLMs for RAG. In molecular tasks, the incorporation of explicit external knowledge can significantly enhance task performance. For a fair comparison, we did not compare with other RAG frameworks.
> 2. **Reference to a Specific Article:**
> We haven't read this article, and the entire project was developed independently without referencing their work. Our experimental cycle was lengthy, and our framework was established before the publication of that article. We consider it a contemporary work. Regarding the better results achieved with the GPT-4 model mentioned in the article, we have two concerns: first, GPT-4 is widely recognized as a leading large model and would undoubtedly interpret data better than our chosen llama3-70B; second, we are concerned that the GPT-4 zero-shot results are much lower than those previously released by MolReGPT, which poses doubts on their prompts; third, considering that GPT-4 has almost exhausted all available text corpora, we are concerned that the ChEBI-20 may have been leaked into the training corpus of the current GPT-4 version. Given these considerations and the fact that we cannot afford the high cost of GPT-4 API, we chose llama3-70B, which, although less effective than claimed in the article, provides more reliable experiments with a lower risk of data leakage. However, we are happy to cite this paper in the revised version.
> 3. **Typographical Error:**
> Sorry for our mistake. We will fix the typo in the revised version.
>
> 4. **Additional Experimental Evidence for Robustness:**
> Thank you for the suggestion. We have added this experiment to illustrate the robustness of our method. The table below shows the comparison of different methods. It can be obvious that our proposed MolReFlect is the most robust method.
>
> | Probing Test | MolT5-base |        | Text+Chem T5-base |        | MolT5-large |        | Text+Chem T5-augm |        | MolReFlect |        |
> |--------------|------------|--------|-------------------|--------|-------------|--------|-------------------|--------|------------|--------|
> |              | ROUGE-2    | METEOR | ROUGE-2           | METEOR | ROUGE-2     | METEOR | ROUGE-2           | METEOR | ROUGE-2    | METEOR |
> | original     | 0.481      | 0.583  | 0.498             | 0.604  | 0.510       | 0.614  | 0.543             | 0.648  | 0.571      | 0.680   |
> | canonical    | 0.315      | 0.450   | 0.381             | 0.515  | 0.390       | 0.532  | 0.377             | 0.514  | 0.416      | 0.543  |
> | hydrogen     | 0.199      | 0.329  | 0.187             | 0.314  | 0.174       | 0.318  | 0.201             | 0.336  | 0.305      | 0.435  |
> | kekulization | 0.333      | 0.475  | 0.413             | 0.574  | 0.405       | 0.546  | 0.410              | 0.546  | 0.443      | 0.569  |
> | cycles       | 0.417      | 0.540   | 0.483             | 0.600    | 0.566       | 0.603  | 0.4575            | 0.581  | 0.545      | 0.658  |

---

### Author Response · Authors · 2024-11-19

We thank all the reviewers for their valuable comments and contributions to the ICLR community. We have responded to the concerns and questions correspondingly below the reviews and updated our paper for you to review.

We sincerely hope that our responses could resolve your questions.

---

### Note · Authors · 2024-11-24

I have read and agree with the venue's withdrawal policy on behalf of myself and my co-authors.